# The p300/YY1/miR-500a-5p/HDAC2 signalling axis regulates cell proliferation in human colorectal cancer

Weimei Tang[1], Weijie Zhou [2], Li Xiang[3], Xiaosheng Wu[1], Pei Zhang[1,4], Jing Wang[1], Guangnan Liu[1], Wenjing Zhang[5], Ying Peng[1], Xiaoting Huang[1], Jianqun Cai[1], Yang Bai[1], Lan Bai[1], Wei Zhu[1], Hongxiang Gu[1], Jing Xiong[1], Chen Ye[1], Aimin Li[1], Side Liu[1,3] & Jide Wang[1,3]

The biological role of miR-500a-5p has not yet been reported in the context of colorectal cancer (CRC). Here, we show that miR-500a-5p expression is decreased in CRC tissues compared with adjacent normal tissues. Low miR-500a-5p expression is associated with malignant progression. Moreover, transfection of CRC cells with miR-500a-5p induces G0/G1 cell cycle arrest and inhibits their growth and migration. Mechanistically, miR-500a-5p directly targets HDAC2 and inhibits HDAC2-mediated proliferation in CRC in nude mice. Furthermore, YY1 binds to the promoter of miR-500a-5p and negatively regulates its transcription. Restoration of miR-500a-5p expression is up-regulated via the p300/YY1/HDAC2 complex. Besides, therapeutic delivery of miR-500a-5p significantly suppresses tumour development in a xenograft tumour model and a HDAC2 inhibitor FK228-treated CRC model. Our studies demonstrate that miR-500a-5p functions as a tumour suppressor in CRC by targeting the p300/YY1/HDAC2 axis, which contributes to the development of and provides new potential candidates for CRC therapy.

[1] Guangdong Provincial Key Laboratory of Gastroenterology, Department of Gastroenterology, Nanfang Hospital, Southern Medical University, Guangzhou 510515, China. [2] Department of Pathology, Nanfang Hospital, Southern Medical University, Guangzhou 510515, China. [3] Department of Gastroenterology, Longgang District People's Hospital, Shenzhen 518172, China. [4] Department of Gastroenterology, Liuzhou General Hospital, Liuzhou, Guangxi 545000, China. [5] Department of Medical Oncology, the First people's Hospital of Yunnan Province, Kunming University of Science and Technology, Kunming 650032, China. Correspondence and requests for materials should be addressed to A.L. (email: lam0725@163.com) or to S.L. (email: liuside@163.com) or to J.W. (email: jidewang55@163.com)

A s one of the major global causes of cancer-related mortality, colorectal cancer (CRC) is surgically curable at early stages, but advanced disease at the metastatic stage is associated with high mortality rates[1]. The overall 5-year cancer-free survival rate was 52.8%, mainly because of the high rates of recurrence and metastasis[2]. Elucidation of the mechanisms underlying CRC tumourigenesis and metastasis will facilitate the search for novel diagnostic biomarkers and the development of effective therapeutic interventions. Over the past 20 years, a number of protein-coding genes that participate in the formation and progression of CRC have been found[3]; however, the function of noncoding RNA, including microRNA (miRNA), remains largely unknown.

miRNAs are small, noncoding RNAs that post-transcriptionally regulate the expression of protein-coding genes by degrading mRNA or terminating translation[4]. Previous studies have shown that miRNAs are aberrantly expressed in many types of cancers and exert tumour-suppressive or oncogenic roles by modulating target gene expression[5,6]. Abnormal expression of these miRNAs have also been reported in CRC carcinoma. These reports suggest that, along with the protein-coding genes, miRNAs may act as a type of important regulator in CRC tumourigenesis[7,8].

miR-500a-5p is a less well-studied miRNA. Several expression profile studies have indicated that miR-500a-5p is dysregulated in liver[9], gastric[10] and breast[11] cancers, and may play an important role in cell proliferation and tumourigenesis. However, its molecular mechanisms and clinical relevance in CRC are not well defined.

Here, we report a suppressive role for miR-500a-5p in CRC cells. Moreover, miR-500a-5p is negatively regulated by its upstream transcription factor YY1, and its expression is modulated via the p300/YY1/ HDAC2 complex. Our results document that miR-500a-5p is able to inhibit tumour development in both xenograft tumours and histone deacetylase (HDAC)2 inhibitor FK228-treated CRC.

## Results

**miR-500a-5p is down-regulated in CRC.** Global miR expression in human normal colon epithelial FHC cells and the human colon cancer cell lines SW620 and LoVo was determined by array analysis using the seventh generation miR Array (Exiqon 208504, Vedbaek, Denmark). Expression levels of 2080 distinct human miRs were examined. Three hundred and fifty-two miRs in LoVo and 324 miRs in SW620 were found to be differentially expressed above the threshold level (1.5-fold) between cancer cells and normal colon epithelial FHC cells and formed the basis for the subsequent analysis. Seventeen miRs were found to share similar expression patterns in both SW620 and LoVo cells. A heat map depicting the two-way hierarchical clustering analysis of these 17 miRs is depicted in Fig. 1a. To confirm these findings, total RNA was harvested from nine cell lines, and quantitative real-time PCR (qPCR) analysis was performed to measure miR-500a-5p levels. As shown in Fig. 1b, these results confirmed that miR-500a-5p levels are significantly decreased in SW480, DLD1, SW1116, SW620, HCT116, LoVo and Caco2 cells compared with the normal human intestinal epithelial FHC and NCM460 cells.

Next, we examined the expression of miR-500a-5p in 81 pairs of human CRC tissues and matched non-tumour tissues. The results revealed that the expression of miR-500a-5p was down-regulated by up to 7.67-fold in 64 of the 81 CRC samples by qRT-PCR (Fig. 1c). Its expression was significantly lower in CRC patient tissues compared with the adjacent normal colon mucosa tissues (Fig. 1d). Moreover, in situ hybridisation (ISH) revealed

that it was localised in both nuclei and cytoplasm of CRC cells, as shown in Fig. 1e and Supplementary Fig. 1.

Furthermore, we divided the patients from TCGA datasets into low-/high-expression groups by the cut-off of the median expression value. The results of the Kaplan–Meier survival curve analysis indicated that 442 patients with cancer of the colon (https://portal.gdc.cancer.gov/projects/) with low miR-500a-5p expression suffered shorter overall survival (Supplementary Fig. 2a; $P = 0.036$) and progression-free survival (Supplementary Fig. 2b; $P = 0.03$). There is no significant difference between the low- and high-level expression of miR-500a-5p with 160 patients with cancer of the rectum (https://xena.ucsc.edu/public-hubs/), but tended to reduce overall survival (Supplementary Fig. 2c; $P = 0.285$) and progression-free survival (Supplementary Fig. 2d; $P = 0.105$).

The above findings suggested that the expression of miR-500a-5p is down-regulated in CRC cells, and a low miR-500a-5p expression was associated with poor survival in CRC patients.

**miR-500a-5p expression attenuates the malignant biological behaviour of CRC.** To determine the clinical relevance of miR-500a-5p expression, we first assessed the clinicopathological features in CRC. No significant association was observed between miR-500a-5p expression and age ($P = 0.436$), gender ($P = 0.752$) or tumour size ($P = 0.291$). However, its expression was significantly correlated with differentiation ($P = 0.007$), lymph node metastasis ($P < 0.001$) and TNM stage (AJCC) ($P < 0.001$) (Supplementary Table 2). These results showed that a low miR-500a-5p expression was associated with malignant progression in CRC patients.

Secondly, we explored the role of miR-500a-5p in the development of CRC. CRC cells with a very low level of miR-500a-5p were transfected with miR-500a-5p mimics, whereas normal human colon epithelial cells were transfected with miR-500a-5p inhibitor. miR-500a-5p expression was confirmed by qRT-PCR (Supplementary Fig. 3a & b). The colony formation assay showed that LoVo cells or SW620 cells transfected with miR-500a-5p mimics yielded significantly fewer colonies compared with those transfected with m-NC (Fig. 1f and Supplementary Fig. 3c). Conversely, the transfection of FHC or NCM460 cells with the miR-500a-5p inhibitor had the opposite effect, resulting in an increased number of colonies compared with cells transfected with i-NC, as predicted (Fig. 1g and Supplementary Fig. 3d).

Similarly, the EdU assay showed that the proliferation rate of colon epithelial cells transfected with miR-500a-5p mimics was significantly decreased (Fig. 1h and Supplementary Fig. 3e), whereas that of those transfected with the miR-500a-5p inhibitor was increased compared with that of control cells (Fig. 1i and Supplementary Fig. 3f). These results indicated that the increased expression of miR-500a-5p in CRC cells resulted in the inhibition of cell proliferation.

Next, we examined the cell cycle profiles by fluorescence-activated cell sorting (FACS) analysis. Indeed, the results revealed an increase in G1-phase cells and a concomitant decrease in S-phase cells among miR-500a-5p mimic-transfected CRC cells (Supplementary Fig. 4a). Cell cycle-related protein expression was assessed by western blotting. Consistent with the accumulation of cells in the G0/G1 phase, the expression of Cyclin D1, CDK4 and CDK6 was significantly decreased in the miR-500a-5p mimic-treated CRC cells, whereas that of Cyclin B1 remained unchanged compared with its expression in the m-NC-treated cells (Supplementary Fig. 4b).

We then evaluated the roles of miR-500a-5p in cell migration and invasion. The wound healing and transwell invasion assay

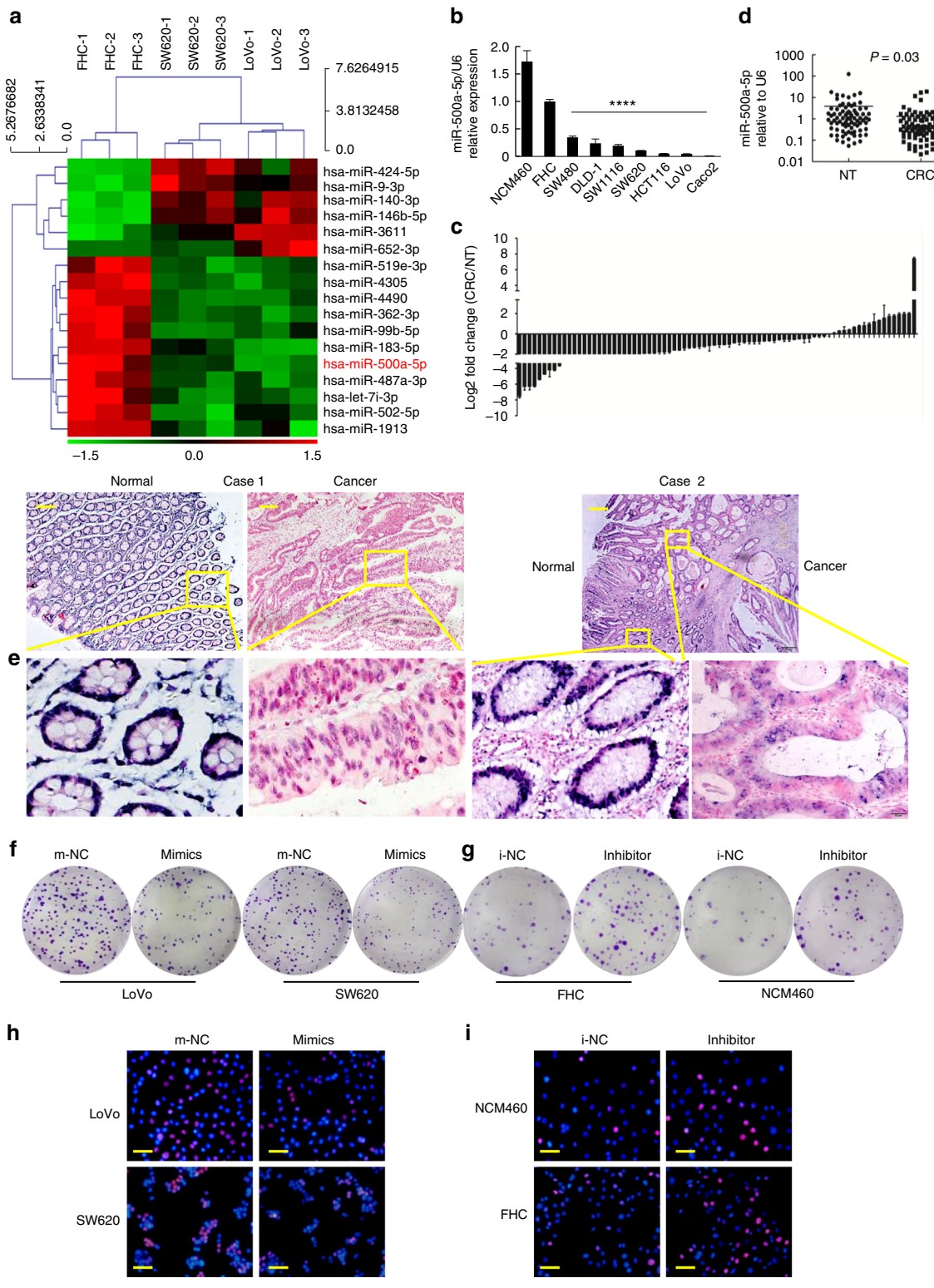

showed that the miR-500a-5p mimics markedly inhibited CRC cell migration and invasion (Supplementary Fig. 5a & b). Consistent with this result, the miR-500a-5p inhibitor significantly increased the cell migration (Supplementary Fig. 5c) and invasion (Supplementary Fig. 5d) capabilities of normal human colon epithelial cells.

The above results suggested that miR-500a-5p suppressed the malignant characteristics of CRC cells.

**HDAC2 is a functional target of miR-500a-5p**. To explore the molecular mechanisms by which miR-500a-5p regulates CRC cell proliferation in vitro, four publicly available bioinformatics algorithms (miRanda, TargetScan, miRTP and RNA22-HSA) and the microarray-based miR-500a-5p signature were used to analyse the target genes of *miR-500a-5p*. The collection of genes with an absolute fold change (FC) > 1.2 in LoVo cells was used to generate an intersection with four published target prediction

**Fig. 1** miR-500a-5p is down-regulated in CRC and associated with malignant biological behaviour. **a** Representative heat map of the miRs that were most differentially expressed in both SW620 and LoVo cells compared with FHC cells. Each row represents an miR and each column represents a cell line. The experiment was performed in triplicate. Red represents up-regulation and green down-regulation, respectively. **b** Validation of miR-500a-5p expression levels in colon epithelial cell lines NCM460, FHC, SW480, DLD1, SW1116, SW620, HCT1116, LoVo and Caco2 cells by qPCR. One-way ANOVA and Dunnett's T3 multiple comparison test. ****$P < 0.001$. **c** Real-time PCR analysis of miR-500a-5p expression in 81 pairs of human CRC tissues and their adjacent normal mucosal tissues. NT, normal mucosal tissues. CRC, colorectal cancer tissue. Error bars represent the mean ± SD from three independent experiments. **d** As analysed by qRT-PCR, miR-500a-5p expression in CRC tissues was significantly lower than that in the corresponding non-cancerous colon mucosa tissue. Student's $t$ test; **$P < 0.05$. **e** Representative ISH images of miR-500a-5p expression. **f** and **g** Effects of miR-500a-5p mimics or inhibitor on the proliferation of colon epithelial cells, as determined by colony formation assay. **h** and **i** DNA synthesis in LoVo cells was measured by the EdU incorporation assay at 48 h after the indicated transfection. Red fluorescence represents the EdU-positive cells; blue fluorescence from the Hoechst 33342 stain represents the total cells. These figures were representative of three to four independent experiments with identical results. Scale bars, 50 μm in (**e**) and 100 μm in (**h**) and (**i**)

engines. The results showed that 66 genes overlapped between the microarray and bioinformatics data (Fig. 2a). Among them, 60 genes, which contained the *HDAC2* gene, were down-regulated in miR-500a-5p-overexpressing cells compared with the control cells (Fig. 2b).

We characterised the correlation between a potential target gene *HDAC2* and *miR-500a-5p* expression; we examined HDAC2 and miR-500a-5p expression in 10 pairs of human CRC tissues and matched non-cancerous colon mucosa by western blot and miRNA analyses. As shown in Fig. 2c and d, cancer tissues (T) exhibited higher HDAC2 protein but lower miR-500a-5p expression levels compared with the corresponding non-cancerous controls (N).

Furthermore, the mRNA level of HDAC2 in the CRC samples obtained from 81 patients was negatively correlated to the miR-500a-5p expression level (Fig. 2e). We then performed a luciferase reporter assay to determine whether HDAC2 was a direct target of miR-500a-5p in CRC cells. The target region sequence of the HDAC2 3′-untranslated region (wild-type (WT) 3′-UTR) or a mutant sequence containing two putative miR-500a-5p sites (MUT1 or MUT2 3′-UTR) was cloned into a luciferase reporter vector (Fig. 3a). Our findings indicated that miR-500a-5p decreased the luciferase activity of the HDAC2-WT 3′-UTR construct after its co-transfection with miR-500a-5p mimics. A mutation in either of the two sites abolished the inhibitory effect of miR-500a-5p on luciferase activity (Fig. 3b).

We further confirmed that *HDAC2* is a target gene of *miR-500a-5p*. The results revealed that the expression of this protein was down-regulated in the CRC cells transfected with miR-500a-5p mimics compared with the m-NC cells but that it was inversely up-regulated in the normal human colon epithelial cells FHC and NCM460 transfected with the miR-500a-5p inhibitor (Fig. 3c).

Next, we determined that whether the effects of miR-500a-5p on CRC cell proliferation and metastasis are indeed mediated by HDAC2. Colony formation, WST-1 and transwell assays indicated that the overexpression of HDAC2 could partly eliminate the inhibition caused by miR-500a-5p (Fig. 3d–f). In addition, the increased proliferation and invasion of FHC and NCM460 cells transfected with the miR-500a-5p inhibitor was significantly rescued by HDAC2 knockdown (Supplementary Fig. 6a & b).

*XIAP* and *RICTOR* were also reported as the target genes of *miR-500a-5p*. Our western blotting data demonstrated that the expression of XIAP or RICTOR is decreased in CRC cell lines SW620 and LoVo treated with miR-500a-5p (Supplementary Fig. 7a), whereas it is increased in the normal human colon epithelial cells FHC and NCM460 transfected with the miR-500a-5p inhibitor (Supplementary Fig. 7b). Our results suggested that XIAP, RICTOR and HDAC2 either independently or cooperatively contribute functionally to the tumour suppressive effects of miR-500a-5p in CRC cells.

Together, these results indicated that miR-500a-5p suppressed CRC development and progression through the down-regulation of HDAC2.

**miR-500a-5p attenuates growth by targeting HDAC2 in CRC cells in vivo**. To assess the effect of miR-500a-5p on tumour growth in vivo, LoVo/m-NC, LoVo/miR-500a-5p, LoVo/Vector, LoVo/HDAC2, and LoVo/miR-500a-5p/HDAC2 cells were implanted subcutaneously into nude mice, and the growth of the resultant primary tumours was monitored (Fig. 4a). The mice injected with LoVo/miR-500a-5p cells developed smaller tumours than those injected with LoVo/m-NC cells. In addition, the mice injected with LoVo/HDAC2 cells developed larger tumours than those injected with LoVo/Vector and LoVo/miR-500a-5p cells, whereas the LoVo/miR-500a-5p/HDAC2 cells formed smaller xenograft tumours than did LoVo/HDAC2 cells (Fig. 4a, b).

We next examined the protein expression of cell proliferation (Ki-67) and angiogenesis (CD105) markers in the xenograft tumours in five groups. Representative images of the tumours after immunohistochemistry (IHC) staining are shown in Fig. 4c and e. The LoVo/miR-500a-5p group exhibited a significantly decreased proliferation rate and tumour vessel density compared with the LoVo/m-NC group (Fig. 4d, f). Moreover, the LoVo/HDAC2 group rapidly proliferated and tumour vessel density increased compared with the LoVo/Vector and LoVo/miR-500a-5p cell groups, whereas LoVo/miR-500a-5p/HDAC2 inhibited the growth rate and tumour vessel density in the LoVo/HDAC2 group (Fig. 4d, f).

These data confirmed that miR-500a-5p down-regulated the HDAC2-mediated growth in CRC in mice.

**miR-500a-5p is negatively regulated by the transcription factor YY1**. Until some years ago, intronic miRNAs were generally thought to be processed from the host gene transcript, with the intronic miRNA and its host gene showing concordant expression levels, since driven by the same promoter[12,13]. However, subsequent studies showed many examples of a poor correlation in the expression levels between miRNAs and their host genes, a phenomenon that could be easily explained by the presence of specific promoters driving the expression of intronic miRNAs[13,14]. *miR-500a-5p* gene is located within the third intron of *CLCN5* gene. Here, we studied the expression relationships between *miR-500a-5p* and its host gene *CLCN5* in 81 CRC tissues. We found that the expression of miR-500a-5p was not related to CLCN5 ($r = -0.159$, $P = 0.157$) (Supplementary Fig. 8).

Therefore, we analysed the 1-kb range of genomic DNA upstream of *miR-500a-5p* gene (http://genome.ucsc.edu/). This experiment showed that the miR-500a-5p promoter (p-Luc-1kb) is able to drive a significantly higher luciferase expression (about 18.2-fold in loVo cells and 15.7-fold in SW620 cells respectively) than the empty vector (pGL3-basic) used as control ($P < 0.001$)

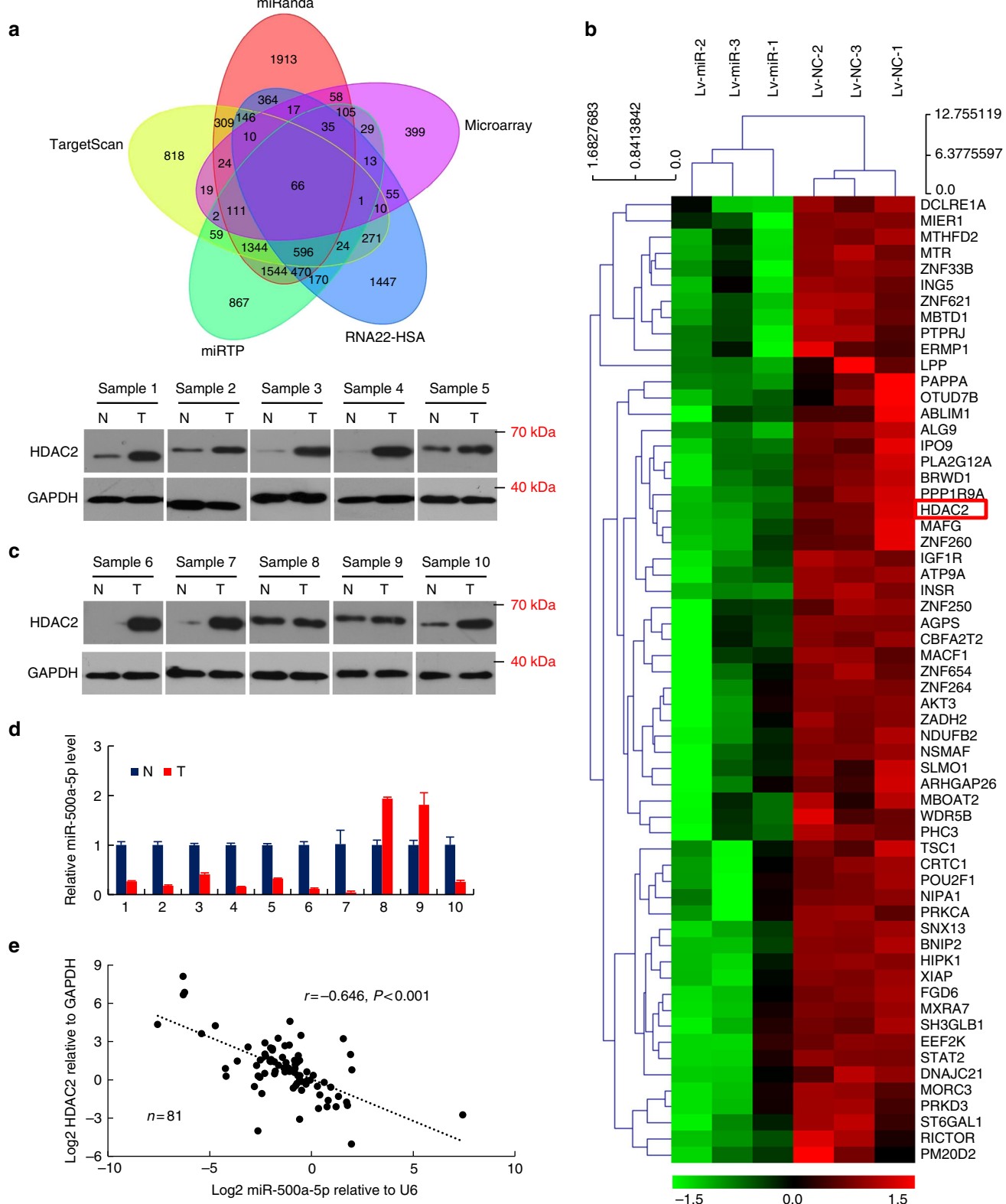

**Fig. 2** miR-500a-5p directly targets HDAC2 in CRC. **a** The five-way Venn diagram indicates the numbers of genes that overlapped in four publicly available bioinformatics algorithms (miRanda, TargetScan, miRTP, RNA22-HSA) and the microarray-based miR-500a-5p signature. **b** The heat map was based on 60 candidate genes that were down-regulated in LoVo cells. Red color represents an expression level above mean, green color represents an expression lower than the mean. **c** and **d** HDAC2 protein and miR-500a-5p expression in ten freshly collected CRC biopsies using western blot and qRT-PCR analyses. **e** In human CRC tissues, HDAC2 was negatively correlated with miR-500a-5p expression. All results were expressed as the mean of three to four independent experiments ± SD. Linear regression analysis. $r = -0.646$, ****$P < 0.001$

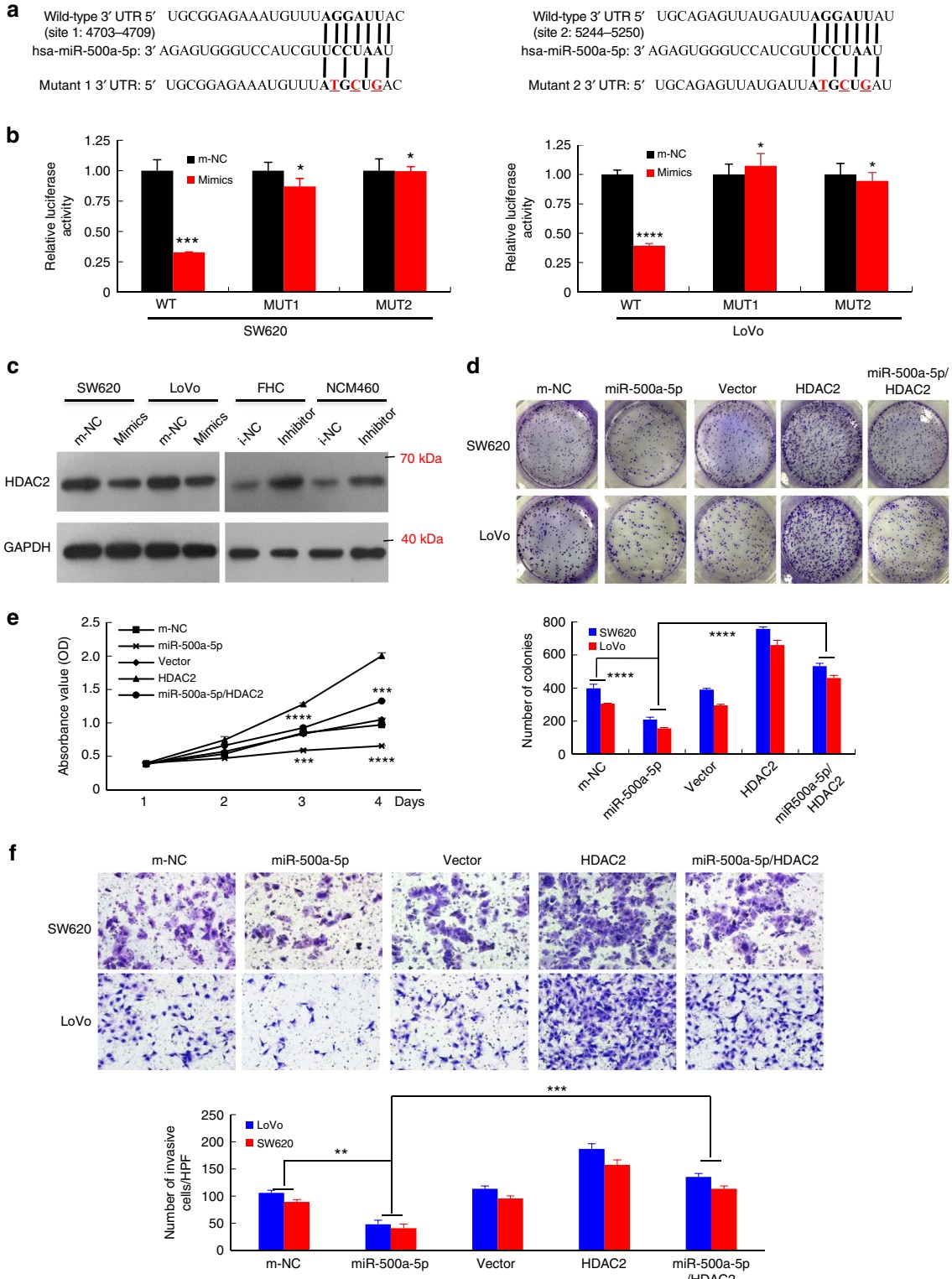

(Supplementary Fig. 9a). Thus, miR-500a-5p expression is driven by its own promoter.

Some studies have shown that the transcription factor YY1 interacts with HDACs[15,16]; thus we investigated whether the transcription factor YY1 could regulate miR-500a-5p in CRC cells. We showed that miR-500a-5p putative promoter region was assessed using the consite software (http://consite.genereg.net/) and the three most probable binding motifs for YY1 lie within the −327 to −333, −622 to −628, and −741 to −747 regions

(Fig. 5a, b). We next cloned the promoter regions of the YY1 site 1, YY1 site 2 and YY1 site 3 of human *miR-500a-5p* upstream of a luciferase gene into a reporter plasmid. The dual-luciferase assay showed that the activity of miR-500a-5p sites 1, 2 and 3 in YY1 cells decreased more than 2.5-fold compared with the vector cells (Fig. 5c and Supplementary Fig. 9b).

To confirm that YY1 could physically bind to the miR-500a-5p promoter in vivo, we performed chromatin immunoprecipitation (ChIP)-qPCR assays in CRC cells expressing exogenous YY1.

**Fig. 3** miR-500a-5p suppresses cell proliferation and invasion by targeting HDAC2 in vitro. **a** The putative miR-500a-5p-binding sequence within the 3′-UTR of HDAC2 mRNA. Mutations in the complementary site for the seed region of *miR-500a-5p* in the 3′-UTR of the *HDAC2* gene are underlined. **b** Reporter plasmids containing either the wild-type 3′-UTR or a mutated 3′-UTR of the *HDAC2* gene were co-transfected into CRC cells with an miR-500a-5p-encoding plasmid, and luciferase activity was measured. WT, wild type; MUT, mutant; site 1, the miR-500a-5p-motif spanning nt 4703–4709; site 2, the motif spanning nt 5244–5250 in the HDAC2 3′-UTR; the data are presented as the mean ± SD of three experiments. Student's *t* test; *$P > 0.05$; *** $P < 0.01$; ****$P < 0.001$. **c** Western blot analysis of HDAC2 protein expression in colon epithelial cells transfected with miR-500a-5p mimics or inhibitor. Cells transfected with a control mimic (m-NC) or inhibitor (i-NC) plasmids were used as a control. **d** Representative results (up) and quantification (down) of crystal violet-stained cell colonies formed by the indicated CRC cells on the 12th day after seeding. Student's *t* test; ****$P < 0.001$. **e** The proliferation of LoVo cells by the WST-1 assay. Student's *t* test; ***$P < 0.01$; ****$P < 0.001$. **f** The invasive activity of cells transfected with an HDAC2 expression plasmid and/or miR-500a-5p mimics was evaluated by the transwell assay. HPF, high-power field. Student's *t* test; **$P < 0.05$ and ***$P < 0.01$. All experiments were repeated at least three times with identical findings

PCR primers located in exon 3 of RPL30 and anti-Histone H3 antibody served as the positive control. The primers were used to amplify the sequence containing the distant upstream region of miR-500a-5p. The distant upstream region was used as the background. The miR-500a-5p promoter region in all the three YY1-binding sites exhibited significant enrichment after immunoprecipitation with an anti-YY1 antibody. No bands were evident in the immunoprecipitates obtained with control IgG (Fig. 5d and Supplementary Fig. 9c).

Then we analysed the expression of YY1 and miR-500a-5p in ten freshly collected CRC biopsies. As shown in Fig. 5e, cancer tissues (T) exhibited a higher YY1 protein expression, whereas miR-500a-5p expression was reduced compared with the corresponding non-cancerous controls (N) (Fig. 2d). YY1 mRNA expression was negatively correlated with miR-500a-5p expression ($P < 0.001$, $r = -0.598$; Fig. 5f).

In addition, a transient expression of YY1 led to a decreased expression not only of the mature form but also of the precursors (pre-miR-500a-5p) or primary transcript (pri-miR-500a-5p) of miR-500a-5p in LoVo and SW620 cells (Fig. 5g and Supplementary Fig. 9d).

We demonstrated that there was a correlation between the clinicopathological parameters of CRC and miR-500a-5p expression (Supplementary Table 2). To explore if the entire YY1/miR-500a-5p/HDAC2 signalling axis was involved, we evaluated the relationship between HDAC2 (Supplementary Table 3) or YY1 (Supplementary Table 4) expression and the same clinicopathological parameters. These data revealed that HDAC2 or YY1 expression was significantly correlated with CRC differentiation ($P = 0.018$ and $P = 0.004$), lymph node metastasis ($P < 0.045$ and $0.001$) and TNM stage ($P = 0.018$ and $P = 0.005$).

Finally, we performed IHC or ISH assay to detect gene expression. The expression of miR-500a-5p in CRC tissues was significantly lower than that in the adjacent normal tissues by ISH. In contrast, HDAC2 and YY1 in CRC tissues were significantly higher than in the adjacent normal tissues by IHC. miR-500a-5p expression was inversely associated with HDAC2 or YY1, as shown in Fig. 5h.

These results suggest that YY1 down-regulates the level of miR-500a-5p and consequently affects the functions of the YY1-miR-500a-5p-HDAC2 pathway in CRC cells.

**Restoration of miR-500a-5p expression is up-regulated via the YY1/p300/HDAC2 complex in CRC cells.** Accumulating evidence suggests that histone acetylation levels are balanced through the opposing activities of histone acetyltransferases (CBP/p300, HATs) and HDACs[17,18]. Previous studies suggest that HDAC2 physically interacts with YY1 and p300 interacts directly with YY1[15]. It has also been reported that HDAC3-p300-YY1[19] or HDAC6-p300-YY1[20] forms a complex and modulates cellular signalling. However, the interaction of HDAC2-p300 and

HDAC2-p300-YY1 complexes in cells remains to be explored. We transfected HA-tagged p300 into CRC cells and assessed whether p300 proteins interact with HDAC2 or YY1. We showed that anti-HA (p300) antibody can co-immunoprecipitate both HDAC2 and YY1 from cell extracts (Fig. 6a). Similarly, HA (p300) or YY1 was co-immunoprecipitated using an anti-HDAC2 antibody (Fig. 6b).

To further confirm HDAC2-p300-YY1 complex formation, we performed co-immunoprecipitation experiments using endogenous proteins. Anti-p300 antibody can co-immunoprecipitate both HDAC2 and YY1 from CRC cell extracts (Fig. 6c). Similarly, both p300 and YY1 were co-immunoprecipitated by an anti-HDAC2 antibody (Fig. 6d), which indicates that these three proteins are likely to be present as a complex in vivo.

Moreover, we screened for other HDAC2 interaction partners by liquid chromatograph-mass spectrometer/mass spectrometer (LC-MS/MS) analysis in LoVo cells. The results indicated that 306 proteins precipitated with HDAC2 antibody were detected when compared to those precipitated with the respective IgGs. A detailed summary of these proteins was given in Supplementary Table 5.

To explore the relationship between miR-500a-5p molecule and the YY1/p300/HDAC2 complex, the CRC cells were transfected with expression plasmids of the vector YY1, HDAC2, and p300, alone or in combination with YY1 + p300, HDAC2 + p300, and YY1 + HDAC2 + p300, respectively, for 48 h. The expression of miR-500a-5p was measured by qRT-PCR. The results revealed that the overexpression of p300 promoted, whereas YY1 or HDAC2 inhibited, the expression of miR-500a-5p. Moreover, p300 co-operation with HDAC2 and/or YY1 could repress the expression of miR-500a-5p in CRC cells (Fig. 6e), indicating an antagonistic effect between YY1 or HDAC2 and p300 to control miR-500a-5p expression.

We have previously shown that the forced expression of YY1 in CRC cells leads to the decreased luciferase activity of the promoter of miR-500a-5p (Fig. 5c). Next, we co-transfected CRC cells with expression plasmids of YY1, HDAC2, and p300, either alone or in combination with YY1 + p300, HDAC2 + p300, and YY1 + HDAC2 + p300, on the reporter gene plasmid pGL3-miR-500a-5p site 1. The results showed that YY1 or HDAC2 alone exerted distinct repressive effects on miR-500a-5p, whereas the ectopic expression of p300 stimulated miR-500a-5p promoter activation (Fig. 6f). An intriguing phenomenon was observed when HDAC2 + p300, YY1 + p300, or HDAC2 + p300 + YY1 were co-transfected, i.e., the decrease in miR-500a-5p transcriptional activity in response to HDAC2 or/and YY1 was moderated by co-transfection with p300 (Fig. 6f).

This finding was confirmed by ChIP studies. We have previously shown that YY1 could physically bind to the miR-500a-5p promoter region in all the three YY1-binding sites (Fig. 5d and Supplementary Fig. 9c). Then, we demonstrate that YY1 binding to the miR-500a-5p promoter region was associated

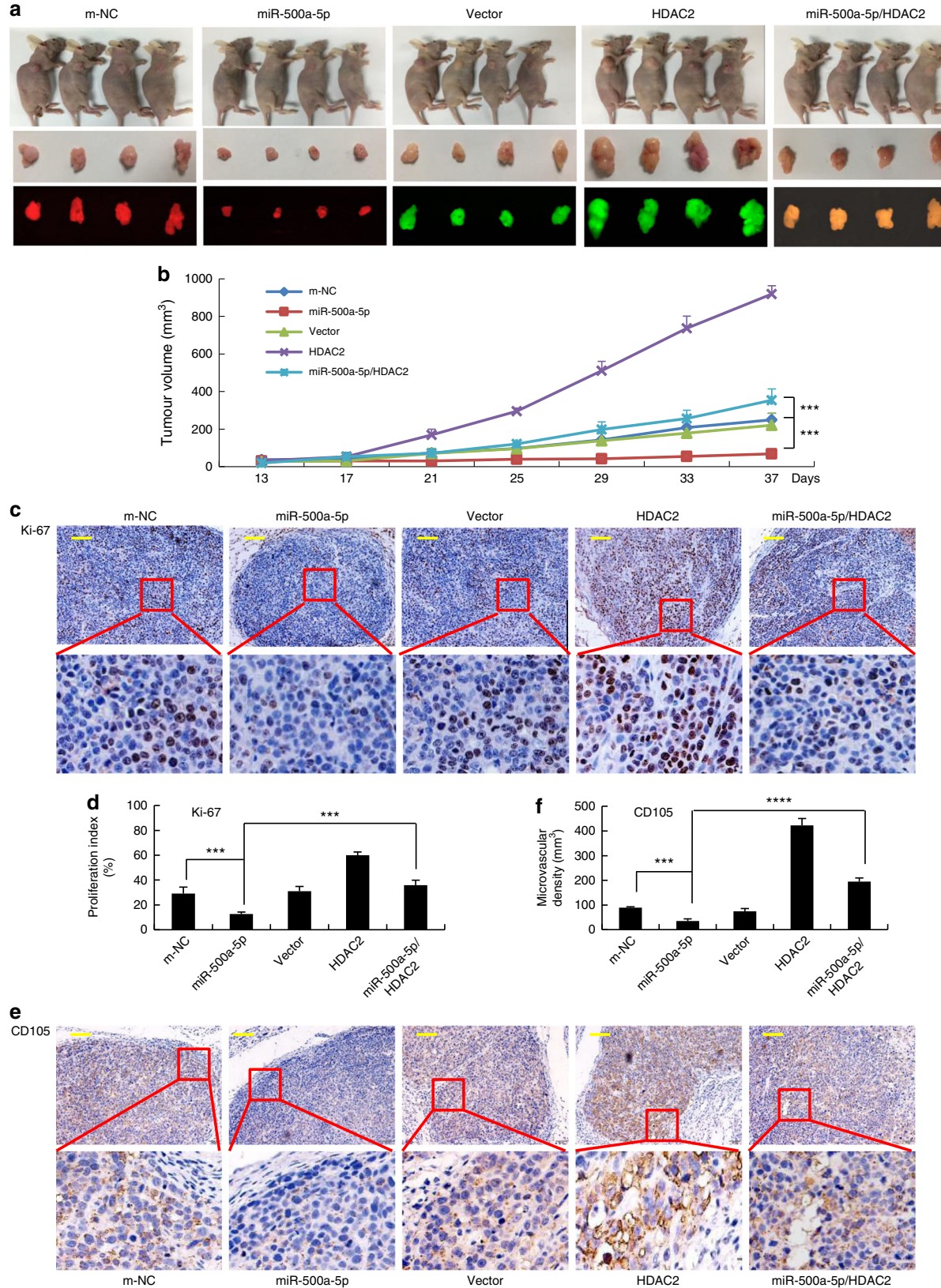

with HDAC2 and p300 in all the three YY1-binding sites in both LoVo and SW620 cells using a re-ChIP assay (Supplementary Fig. 10a, b).

Next, the overexpression of YY1 increased the binding of YY1 to the miR-500a-5p site 1, 2 and 3 proximal promoter. Moreover, a transient transfection of p300 decreased, whereas the

overexpression of HDAC2 increased, the binding of YY1 to the miR-500a-5p promoter in CRC cells (Fig. 6g and Supplementary Fig. 11a).

In contrast, small interfering RNA (siRNA)-mediated knock-down of YY1 decreased the binding of YY1 to the miR-500a-5p promoter and vice versa (Fig. 6h and Supplementary Fig. 11b). In

**Fig. 4** miR-500a-5p modulates CRC tumour growth by targeting HDAC2 in vivo. **a** Fluorescence images of subcutaneous tumours of mice injected with LoVo/m-NC, LoVo/miR-500a-5p, LoVo/Vector, LoVo/HDAC2, and LoVo/miR-500a-5p/HDAC2 cells. The growth of the resultant primary tumours was monitored. **b** Tumour size was measured starting from the 13th day after tumour cell inoculation in each group. Student's $t$ test; ***$P < 0.01$, m-NC vs miR-500a-5p and ***$P < 0.01$, miR-500a-5p vs miR-500a-5p/HDAC2. **c–f** Immunohistochemical (IHC) detection and quantification of Ki-67 and CD105 protein expression in subcutaneous tumours from mice injected with LoVo cells. Student's $t$ test; ***$P < 0.01$, m-NC vs miR-500a-5p and ***$P < 0.01$, miR-500a-5p vs miR-500a-5p/HDAC2 in Ki-67. ***$P < 0.01$, m-NC vs miR-500a-5p and ****$P < 0.001$, miR-500a-5p vs miR-500a-5p/HDAC2 in CD105. Scale bars, 200 μm in (**c**) and (**e**)

addition, the ectopic expression of p300 weakened HDAC2 to bind to the miR-500a-5p promoter YY1-binding sites in CRC cells (Supplementary Fig. 11c & d).

These results suggested that the overexpression of miR-500a-5p was up-regulated through the YY1/p300/HDAC2 complex in CRC cells.

**Therapeutic delivery of miR-500a-5p is able to inhibit tumour development in both xenograft tumours and FK228-treated CRC.** To investigate the molecular mechanism underlying the preventive effects of miR-500a-5p in CRC, we performed an apoptosis assay. As shown in Fig. 7a, miR-500a-5p induced greater apoptosis than m-NC in LoVo cells by FACS analysis. To evaluate the role of miR-500a-5p in the susceptibility of cells to HDAC inhibitor FK228-mediated apoptosis, LoVo cells were treated with FK228 (romidepsin, 10 nM, using dimethyl sulfoxide as the vehicle) for 48 h in vitro. As shown in Fig. 7a, the apoptotic cells of the cells transfected with miR-500a-5p + FK228 were significantly increased relative to miR-500a-5p by flow cytometric analysis. Similarly, apoptotic induction was further confirmed by Hoechst 33342 staining at the single-cell level (Fig. 7b). These findings suggested that miR-500a-5p promoted the apoptosis and enhanced the susceptibility of cancer cells to apoptotic triggers induced by FK228 in vitro.

To investigate the therapeutic potential of miR-500a-5p in CRC, we developed a xenograft mouse model and an FK228-treated CRC model. We subcutaneously injected LoVo cells into the right flanks of nude mice of both the models. When tumour nodules became visible (~3–5 mm in diameter, 11 days), FK228 was intraperitoneally injected as a co-treatment. The tumour sizes were then monitored every 4 days (Fig. 7c). As shown in Fig. 7d, the tumour volumes of the miR-500a-5p- and miR-500a-5p + FK228-treated mice were markedly smaller than those of the m-NC-treated mice 39 days after injection of the LoVo cells.

Tumours injected with a combination of miR-500a-5p and/or FK228 exhibited the greatest percentage of apoptotic cells by TdT-mediated dUTP Nick-End Labeling (TUNEL) staining. In contrast, tumours injected with m-NC exhibited relatively few TUNEL-positive cells (Fig. 7e). These findings suggest that miR-500a-5p enhances the susceptibility of cancer cells to apoptotic triggers that are induced by FK228, an HDAC inhibitor.

These data demonstrate that miR-500a-5p acts as a tumour suppresser and has therapeutic potential for the treatment of CRC.

## Discussion

Accumulating evidence indicates that miRNAs contribute to cancer pathogenesis. As miRNAs may serve as tumour suppressors or oncogenes, dysregulation of miRNAs is associated with cancer initiation and progression[5,6,21]. In the current study, miR-500a-5p was down-regulated in CRC tissues. Ectopic miR-500a-5p expression remarkably inhibits CRC cell proliferation and migration by targeting HDAC2. Moreover, the expression of miR-500a-5p was down-regulated via the YY1/p300/HDAC2

transcription complex in CRC cells. In addition, miR-500a-5p induced apoptosis and sensitised cancer cells to HDACi, FK228, both in vitro and in vivo. These data suggested that miR-500a-5p might serve as a novel biomarker or therapeutic target for CRC.

Hsa-miR-500a-5p (known as miR-500a) was first identified in human hepatocellular carcinoma. Sanchez Freire V et al. found that miR-500a-5p may mediate the down-regulation of neurokinin-1 receptor in chronic bladder pain syndrome[22]. Studies have reported miR-500a-5p dysregulation in several cancers, including liver cancer and gastric cancer[9–11]. However, the function of miR-500a-5p in CRC has remained unknown. Here, we showed that miR-500a-5p expression was down-regulated in human CRC tumour tissues and cells. Moreover, its expression was significantly correlated with adverse clinicopathological factors such as differentiation, lymph node metastasis, and TNM stage. These strong correlations suggest that ectopic expression of miR-500a-5p may inhibit tumour cell proliferation, cell cycle, invasion and metastasis. The results of this study confirm the tumour-suppressive role of miR-500a-5p in CRC cells and provide evidence of the potential utility of this miRNA for miRNA-based cancer therapy.

HDACs are frequently overexpressed in a broad range of cancer types, where they alter cellular epigenetic programming to promote cell proliferation and survival[23–25]. The class I HDACs are the most frequently overexpressed in cancers, particularly HDACs 1, 2 and 3. Recent progress has highlighted the significance of HDAC2 in tumour progression[26,27]. Previous studies have shown that HDAC2 expression was up-regulated in CRC cancer compared with its matched normal tissues[28]. In this regard, an enforced miR-183 expression triggered apoptosis and inhibited anchorage-independent colony formation in vitro and xenograft growth in mice. HDAC2 overexpression reduced miR-183 levels and counteracted the induction caused by HDAC2 depletion or HDAC inhibitor treatment[29]. Consistently, we identified that HDAC2 was a direct functional target of miR-500a-5p in CRC. First, the miR-500a-5p level was inversely associated with HDAC2 protein. Second, ectopic miR-500a-5p expression markedly reduced the activity of a luciferase reporter containing the 3′-UTR sequence of HDAC2. Third, the regulation of miR-500a-5p expression could significantly reverse the inhibitory effects of HDAC2. Fourth, the restoration of miR-500a-5p in HDAC2-overexpressing CRC cells strongly inhibited cell proliferation and metastasis both in vitro and in vivo. The data demonstrate the new role of miR-500a-5p as an important post-transcriptional regulator of HDAC2 in CRC cells.

YY1 is a 65-kDa member of the GLI-kruppel family of zinc finger transcription factors, which regulate various developmental and differentiation processes. Most processes mediated by YY1 are cancer-related, strongly implicating its importance in cancer development and progression. Indeed, overexpression of YY1 has been observed in various types of cancers[30–32]. A recent survey examining the number of promoters that contain potential YY1-binding sites provided overwhelming results, and the reports addressing the number of promoters that can be regulated by YY1 are equally remarkable[33,34]. Consistently, we showed that YY1

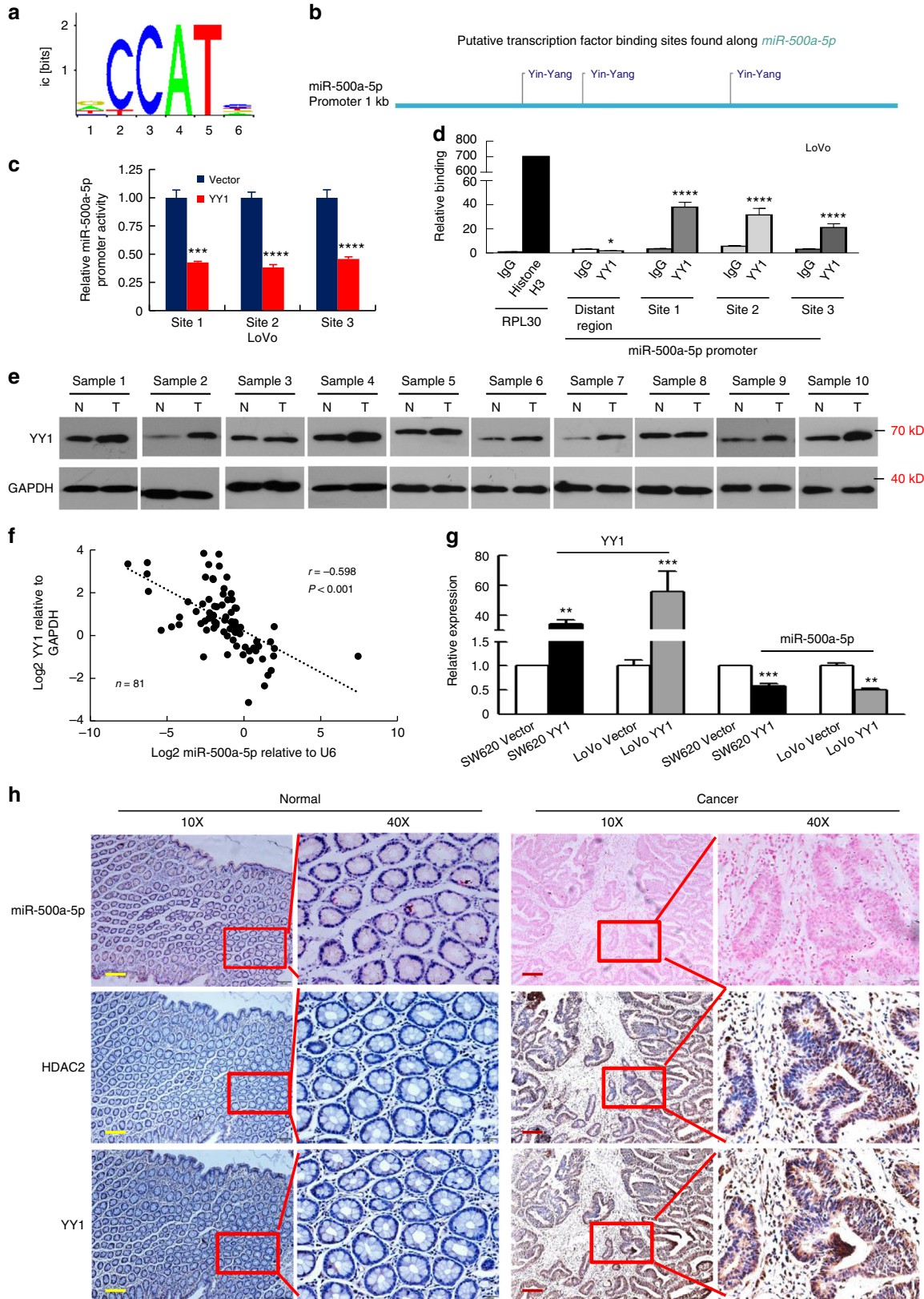

was recruited and bound to the promoter region (−1 to −1kb) of miR-500a-5p in the bioinformatics analysis. Subsequent experiments confirmed that YY1 could negatively regulate miR-500a-5p expression by directly binding to its promoter region. Thus, YY1 is a novel transcriptional repressor of miR-500a-5p expression.

It was previously reported that p300 is mutated in several forms of cancer, suggesting a tumour suppressor role for this protein and providing co-repressor function. Some reports have shown that p300 exists in multi-molecular complexes in vivo and functions as a co-activator or co-repressor for a variety of

**Fig. 5** miR-500a-5p is regulated directly by the transcription factor YY1. **a** The transcriptional factor YY1-binding motif was predicted by informatics analysis. **b** Schematic illustration of the miR-500a-5p promoter with three potential YY1-binding sites. **c** Luciferase activity of the miR-500a-5p promoter construct after the transfection of the YY1 plasmid in LoVo cells. Student's *t* test; ***$P < 0.01$ and ****$P < 0.001$. **d** A ChIP-qPCR assay demonstrated the direct binding of YY1 to the miR-500a-5p promoter in LoVo cells. Student's *t* test; *$P > 0.05$ and ****$P < 0.001$. Gene enrichment was quantified relative to input controls by qPCR using primers specific for the promoter regions of *miR-500a-5p*. Results are shown as a fold change of qPCR value over IgG. **e** YY1 protein expression in ten freshly collected CRC biopsies using western blot analysis. **f** YY1 was negatively correlated with miR-500a-5p expression in human CRC tissues measured by qPCR. Linear regression analysis, $r = -0.598$; ****$P < 0.001$. **g** Expression of YY1 and mature miR-500a-5p in YY1-overexpressing CRC cells by real-time PCR. Student's *t* test; **$P < 0.05$ and ***$P < 0.01$. All experiments were repeated three times with identical findings. **h** ISH analysis of miR-500a-5p and IHC analysis of HDAC2 and YY1 were performed. These figures are representative of colorectal tissues from 10 cancerous and 10 non-cancerous patients. The scale bars represent 200 μm

genes[19,35,36]. Consistently, we showed that p300 co-operates with transcription factor YY1 and HDAC2 to regulate the miR-500a-5p promoter. YY1 and HDAC2 inhibit miR-500a-5p promoter transcription. Moreover, the ectopic expression of p300 weakened YY1 and HDAC2 to bind to the miR-500a-5p promoter YY1-binding sites and thus activated the miR-500a-5p promoter transcription indeed in CRC cells. Therefore, we believe that the p300/YY1/miR-500a-5p/HDAC2 signalling axis plays important roles in cancer development.

FK228 was originally isolated from *Chromobacterium violaceumas*[37]. It is a novel antitumour depsipeptide that inhibits *HDAC2* and restores the expression of genes aberrantly suppressed in cancer cells[38]. We identified that miR-500a-5p functions as a tumour suppressor, and we wondered whether this miRNA combined with HDAC2 inhibitor, FK228, induced apoptosis in CRC cells. Data from both in vivo and in vitro analyses showed miR-500a-5p mimics induced apoptosis in CRC cells. Moreover, the results from an FK228-induced CRC cell model demonstrated that the therapeutic delivery of miR-500a-5p could enhance FK228-induced apoptosis. Thus, it provides a wider perspective on CRC intervention/prevention and treatment.

In summary, our study shows that miR-500a-5p is significantly down-regulated in CRC and suppresses CRC tumour growth and metastasis by targeting HDAC2. Furthermore, miR-500a-5p is negatively regulated by the transcription factor YY1. Moreover, miR-500a-5p expression is elevated via the YY1/p300/HDAC2 complex. In addition, the ectopic overexpression of miR-500a-5p and HDACi induces apoptosis in CRC cells both in vitro and in vivo. The proposed signal cascade for miR-500a-5p in regulating cell proliferation and apoptosis is schematically shown in Fig. 8. Therefore, our results suggest that the p300/YY1/miR-500a-5p/HDAC2 axis may be a potent therapeutic approach for CRC.

## Methods

**Cell culture, chemicals and reagents**. Non-tumourigenic immortalised adult human colon epithelial cell lines, FHC and NCM460, and colon cancer cell lines SW620, LoVo, SW480, SW1116, HCT116, DLD1 and Caco2 cells were obtained from ATCC. The cells were grown in RPMI 1640 containing 10% foetal bovine serum (Life Technologies, Monza, Italy), 1% glutamine (Life Technologies) and 1% penicillin/streptomycin (Life Technologies) in a humidified incubator at 37 °C with an atmosphere of 5% $CO_2$.

FK228 (Romidepsin) was purchased from Sigma (St. Louis, MO, USA). Rabbit antibodies against histone H3 (D2B12) was purchased from Santa Cruz Biotechnology (Santa Cruz, CA, USA).

**RNA isolation and real-time qRT-PCR**. Total RNA was extracted from tissues and cells using Trizol reagent (Invitrogen) and reverse-transcribed using an M-MLV RT Kit (Promega). For the detection of mature, precursors and primary- miR-500a-5p, specific RT primers and PCR primers (Gene Copoeia, San Diego, CA, USA) were constructed, and cDNA was amplified using SYBR® Green PCR Master Mix (Toyobo, Osaka, Japan). The relative expression level of mRNA or miRNA was normalised to glyceraldehyde-3-phosphate dehydrogenase (GAPDH) or snRNA

U6 expression and was calculated using the $2^{-\Delta\Delta Ct}$ method. The primers are shown in Supplementary Table 1.

**Microarray**. MicroRNA expression profiles of FHC group, SW620 and LoVo group ($n = 3$ per group) were generated using the miRCURYHy3/Hy5 power labelling kit and the miRCURY LNA Array (v.10.0; 757 human miRs) by Exiqon (KangChen, China). The expression values are log2 (Hy3/Hy5) ratios. The differentially expressed microRNAs were analysed above the threshold level (1.5) between cancer cells and normal colon epithelial FHC cells. Unsupervised hierarchical clustering of miRNAs was performed. Microarray data are deposited in the NCBI database (Gene Expression Omnibus, GEO database accession no: GSE115108).

Total RNAs of LoVo cells stably transfected with LV-miR-500a-5p and LV control (NC) were isolated by Trizol reagent (Invitrogen) according to the manufacturer's instructions, and the whole-genome expression microarray (Affymetrix) was performed by Shanghai Biochip Corp. The genes exhibited differential expression patterns between the LoVo/*miR-500a-5p* cells and i-NC cells, and the number of genes had an absolute FC greater than 1.2. Raw and processed data from the microarray were deposited in NCBI's GEO database under the accession number GSE122884.

**Patients and specimens**. Eighty-one CRC tissues and 81 adjacent non-tumour tissues collected from CRC patients between January 2016 and June 2016 were selected from the Department of Surgery of Nanfang Hospital, Southern Medical University, China. The Ethics Committee of the Southern Medical University approved our experimental protocols. All CRC cases were confirmed by a senior pathologist and staged based on the 2011 Union for International Cancer Control TNM classification of malignant tumours. The pathological diagnoses of all enrolled patients were confirmed by two different pathologists according to the WHO grading system[39].

**In situ hybridisation and immunohistochemistry analyses**. ISH analysis was performed on paraffin-embedded tissue sections. A double digoxigenin (DIG)-labelled mercury-locked nucleic acid probe was used as the miR-500a-5p probe [miRCURY LNA™ detection probes (Exiqon, Vedbaek, Denmark)]. The probe sequence was as follows: 5′-TCTCA CCCAG GTACRC AAGGA TTA-3′. In brief, sections (4 μm) of archived paraffin-embedded specimens were deparaffinised in xylene and then rehydrated through an ethanol dilution series (99.9–70%). Sections were treated with Pepsin for ISH (BOSTER, Wuhan, China) at 37 °C for 10 min and then dehydrated through an ethanol dilution series (70–99.9%). Slides were incubated in a DIG-labelled probe diluted to 40 nM in hybridisation buffer at 50 °C for 2 h. Stringent washes were performed with 5 × SSC, 1 × SSC and 0.2 × SSC buffers at 50 °C for 10 min each. The samples were incubated with DIG blocking reagent (Roche) in maleic acid buffer at 30 °C for 15 min and alkaline phosphatase-conjugated anti-digoxigenin (diluted 1:500 in blocking reagent, Roche) at room temperature for 60 min. Enzymatic development was performed using 4-nitro-blue tetrazolium and 5-brom-4-chloro-30-indolyl phosphate substrate (Roche), which forms dark blue 4-nitro-blue tetrazolium-formazan precipitates at 30 °C overnight, followed by nuclear fast counterstaining for 2 min. The slides were then immersed in water, dehydrated in alcohol solutions and mounted with Eukitt mounting medium (VWR, Herlev, Denmark). Scrambled probe was used as a control.

IHC analysis was conducted to determine HDAC2 (1:100, A2084; ABclonal), YY1 (1:100, 66281-1-lg; Proteintech), Ki-67 (1:4000, 27309-1-AP; Proteintech) and CD105(A-8) (1:100, ab169545; Abcam) protein expression in CRC[3,40]. Briefly, paraffin-embedded tissue blocks were cut into 5-mm sections and transferred to glass slides. The slides were deparaffinised with xylene, rehydrated with ethanol, washed, and subjected to microwave retrieval in a citrate buffer. Sections were then immersed in 3% hydrogen peroxide to block endogenous peroxidase activity and incubated with the first antibodies followed by incubation with the biotin-linked anti-Rabbit IgG (Dako, Copenhagen, Denmark) in combination with the diaminobenzidine (DAB) complex. Normal rabbit or mouse IgG (Sigma) was used as the isotype controls.

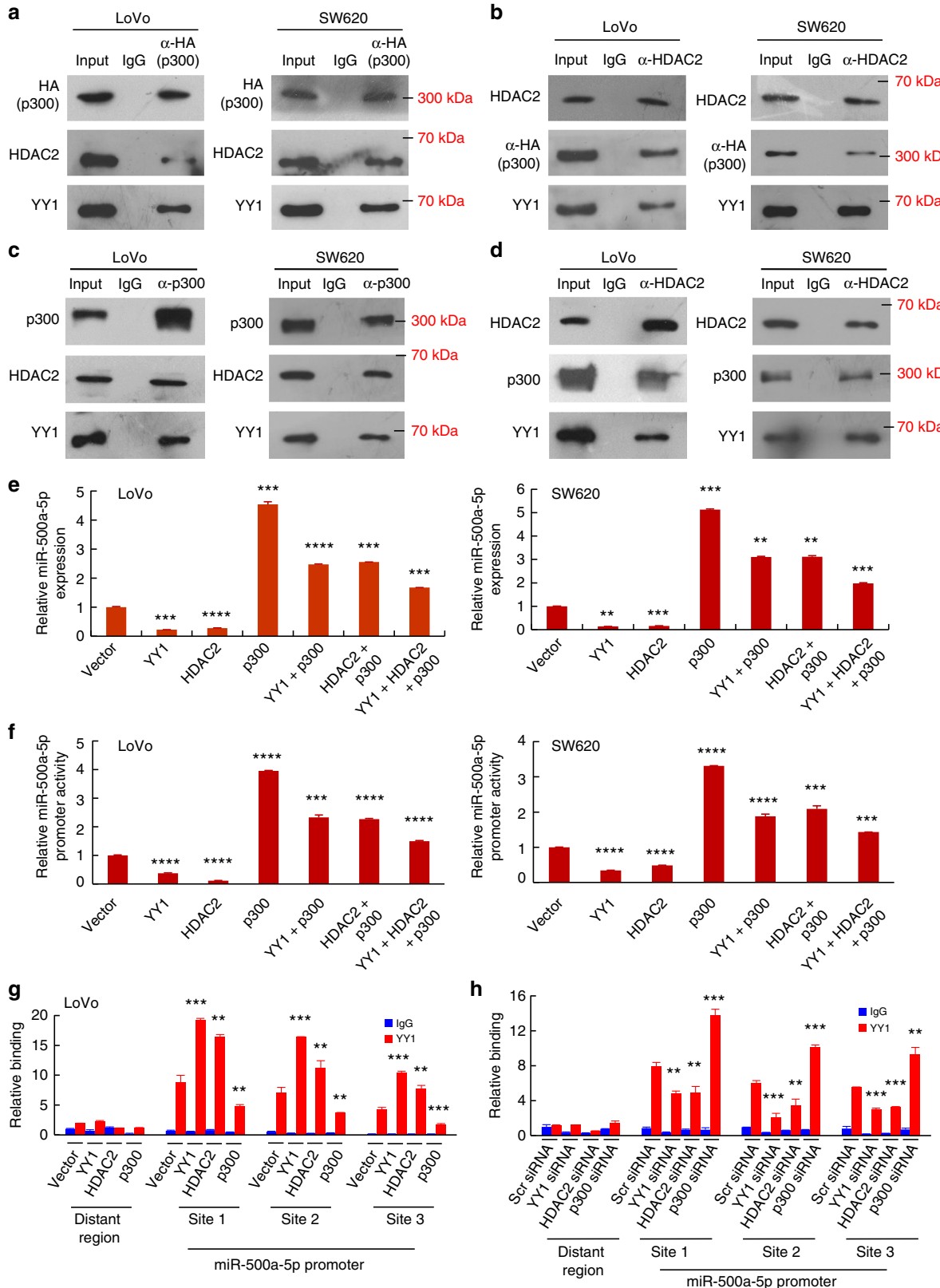

**Bioinformatics**. Potential miRNA targets were predicted and analysed using microarray data and four of the following publicly available algorithms: RNA22-HSA (https://cm.jefferson.edu/rna22/), miRTP (https://genome.tugraz.at/MiRTP/), TargetScan (http://www.targetscan.org/) and miRanda (http://www.microrna.org/microrna/home.do). The targets were accepted only when they were positive in all five analyses.

**Oligonucleotide transfection**. An miR-500a-5p mimic (40nmol/l/well; sense 5'-UAAUCCUUGCUACCUGGGUGAGA-3', antisense 5'-ACGUGA CACGUUCGGAGAATT-3'; miR-500a-5p inhibitor (100nmol/l/well, 5'-UCU CACCCAGGUAGCAAGGAUUA-3'; GenePharma Co., Ltd, Shanghai, China) or their corresponding controls (m-NC for mimics, sense 5'-UUCUCCGAACGU GUCACGUTT-3', antisense 5'-ACGUGACACGUUCGGAGAATT-3' and i-NC

**Fig. 6** miR-500a-5p expression is up-regulated via the YY1/p300/HDAC2 complex. **a** In vivo co-immunoprecipitation assays showing the presence of a complex containing YY1, HDAC2 and p300. HA-tagged p300 plasmid was transfected into CRC cells. Immunoprecipitation was performed with anti-HA (p300) antibody, and pre-immune normal mouse immunoglobulin G (nm IgG) was used as a control. Western blot analysis was performed with an anti-HDAC2 (α-HDAC2) and a YY1 antibody. **b** HDAC2 plasmid was transfected into CRC cells. Cell lysates were immunoprecipitated using an anti-HDAC2 antibody or the control antibody normal mouse immunoglobulin G (nm IgG). Western blotting was performed with anti-HA (p300) and YY1 antibodies. **c** and **d** Whole cell lysates were immunoprecipitated using an anti-p300 (**c**) or anti-HDAC2 (**d**) antibody or the control antibody normal nm IgG. Western blotting was performed with HDAC2 or p300 and YY1 antibodies. **e** Gene expression studies by qRT-PCR in SW620 and LoVo cells. Student's $t$ test; **$*P < 0.05$, ***$P < 0.01$ and ****$P < 0.001$ between vector and gene. **f** The miR-500a-5p promoter construct was co-transfected with vector, YY1, HDAC2, and p300 either alone or in combination with HDAC2 + p300, YY1 + p300, and HDAC2 + p300 + YY1 in CRC cells. Luciferase assays were performed. Values are means ± SD ($n = 3$). Student's $t$ test; ***$P < 0.01$ and ****$P < 0.001$ between vector and gene. **g** and **h** Binding of the transcription factor YY1 to miR-500a-5p was analysed with anti-YY1 ChIP using qPCR. Student's $t$ test; **$*P < 0.05$ and ***$P < 0.01$ between vector and gene. All figures are representative of three independent experiments

for inhibitor, 5'-CAGUACUUUUGUGUAGUACAA-3') were transfected using Lipofectamine 2000 transfection reagent (Invitrogen, Foster City, CA, USA) according to the manufacturers' protocols. The transfection efficiency was evaluated by RT-PCR.

**DNA constructs and siRNA.** pENTER-HDAC2 and YY1 or p300 and empty vector (pENTER) plasmids encoding a FLAG or HA tag were purchased from Vigene (Rockville, MD, USA). Ablation of HDAC2, YY1 or p300 was performed by transfection with siRNA duplex oligos, which were synthesised by Genepharma Company (Shanghai, China). The sequences of the siRNAs were as follows: (sense strand) Scrambled siRNA, HDAC2 siRNA: 5'-AAGCATCAGGATTCTGTTA-3', YY1 siRNA: 5'-CGAGGATCAGATTCTCATC-3'[41], and p300 siRNA: 5'-AACCCCUCCUCUUCAGCACCA-3'. Cell transfection was performed with Lipofectamine TM 2000 (Invitrogen) as described in the manufacturer's protocol.

**EdU incorporation assay.** Each group of isolated tumour cells was seeded onto 96-well plates in triplicate at a density of $10^3$/well and incubated for 48 h. Subsequently, the cells were incubated for an additional 2 h in the respective media containing 50 μM EdU (RiboBio, Guangzhou, China). Cell proliferation was detected using a Cell-Light™ EdU Cell Proliferation Detection Kit (RiboBio, China) following the manufacturer's instructions. DNA was incubated with Hoechst 33342 stain (100 μl/well) for 30 min and visualised using an inverted fluorescence microscope (Leica DM5500, Germany). For each EdU experiment, five random fields were imaged at 100 × magnification. Captured images were processed and analysed using ImageJ software. The number of EdU-positive cells was determined by Hoechst nuclear staining and expressed as a percentage of the total number of cells in each field.

**Protein extraction and western blotting.** Proteins were extracted from sub-confluent cultures of cells and then characterised by western blot analysis. Cells were lysed in lysis buffer, resolved on a sodium dodecyl sulphate-polyacrylamide electrophoresis gel, and transferred onto an Immobilon-P membrane (Millipore, Billerica, MA, USA). The membrane was blocked with 5% non-fat milk in phosphate buffered saline containing 0.05% Tween (PBST) for 1 h at room temperature, and then probed with a primary antibody overnight at 4 °C. After extensive washing, the membrane was incubated with a secondary antibody conjugated to horseradish peroxidase (1:10,000; Santa Cruz Biotechnology Inc.) for 1 h at room temperature. Blots were developed using ECL (PE LifeScience, Waltham, MA, USA).

The following antibodies were used: CDK4 (DCS-35) (1:2000, sc-23896; Santa Cruz), CDK6 (B10) (1:1000, sc-7961; Santa Cruz), Cyclin B1 (G11) (1:200, sc-166757; Santa Cruz), Cyclin D1 (A12) (1:200, sc-8396; Santa Cruz), HDAC2 (C-8) (1:200, sc-9959; Santa Cruz), GAPDH (G-9) (1:3000, sc-365062; Santa Cruz), XIAP (1:1000, 10037-2-lg; proteintech), RICTOR (1:1000, 27248-1-AP; proteintech), p300 (3G230/NM-11) (1:1000, ab14984; Abcam), and YY1 (1:5000, 66281-1-lg; proteintech). Uncropped western blot scans from main blots are displayed in Figs. 2c, 3c, 5e, 6, and Supplementary Figs. 4b and 7.

**Cell proliferation assays.** Cell proliferation assays were performed using the WST-1 assay (Roche Molecular Biochemicals, Basel, Switzerland). Cells were plated in 96-well plates at a density of $1 × 10^4$ cells per well and cultured in a growth medium. At the indicated time points, the number of cells in triplicate wells was measured based on the absorbance at 450 nm using an automatic plate reader.

**Colony formation assay.** Cells were seeded in flat-bottomed twelve-well plates with 1 ml RPMI 1640 supplemented with 10% foetal bovine serum. Two days later, the medium was replaced with a new medium and the culture was continued for an additional 12 days. Thereafter, the colonies were fixed with methanol and stained with 0.05% crystal violet. The colonies were counted directly under a Zeiss microscope.

**Luciferase activity assay for the 3'-UTR study.** The luciferase reporter plasmid carrying the WT or mutated (MUT) HDAC2 3'-UTR (pMIR-report-HDAC2-3'-UTR and pMIR-report-HDAC2-MUT1-3'-UTR, pMIR-report-HDAC2-MUT2-3'-UTR, respectively) was transfected into CRC cells along with the miR-500a-5p mimics using Lipofectamine 2000 (Invitrogen). After transfection (36–48 h), the cells were lysed, and luciferase activity was measured with the Dual-Luciferase Reporter Assay system (Promega, Madison, WI, USA). The sequences of HDAC2-WT-3'-UTR and HDAC2-MUT-3'-UTR are shown in Fig. 3a.

**Promoter analysis.** The 1-kb region directly upstream of miR-500a-5p was predicted using UCSC software. Analysis of YY1-binding sites on the miR-500a-5p promoter was performed using the TF prediction programme Consite (http://consite.genereg.net/). miR-500a-5p promoter (miR-500a-5p-p) construct contained the YY1-binding sites 1 (miR-500a-5p-p site 1:−333 ~ −327), 2 (miR-500a-5p-p site 2: −628~ −622), 3 (miR-500a-5p-p site 3: −747 ~ −741) or 1 kb (p-Luc-1 kb, −1003). Dual-luciferase assay was performed using the Dual-luciferase Reporter Assay kit (Promega, Madison, WI, USA). Briefly, $1 × 10^5$ cells were seeded in each well of a 24-well tissue culture plate. The cells were incubated until 70% confluent. Cells in each well were transfected with 0.8 μg pGL3 basic vector or the pGL3 vector harbouring the various miR-500a-5p promoter regions using Lipofectamine 2000 reagent. The Renilla luciferase reporter pRL-CMV plasmid (Promega), 0.01 μg per well, was co-transfected as the internal control. After transfection for 4 h, cells were transferred into a normal medium. To examine the relationship between YY1 and miR-500a-5p promoter activity, miR-500a-5p site 1, 2 or 3 reporter plasmid was co-transfected with YY1 or vector. Forty-eight hours later, the cells were treated with passive lysis buffer. Luciferase activities were measured with a luminometer (lumat LB9507, Berthold, Bad Wildbad, Germany). The firefly luciferase activity value was normalised to the renilla activity value. Promoter transcription activity was presented as the fold induction of relative luciferase unit (RLU) compared with basic pGL3 vector control (the RLU was the value of the firefly luciferase unit divided by the value of the renilla luciferase unit.) The oligonucleotide primers used in the luciferase activity assays are listed in Supplementary Table 1.

**ChIP and Re-ChIP.** ChIP assays were performed according to the protocol provided with the Chip assay kit (Upstate Cell Signalling Solutions, Lake Placid, NY, USA). The lysates were incubated either with antibodies specific for YY1 or with normal mouse IgG. The PCR generated 194-, 180- and 180-bp products from the miR-500a-5p proximal (<1000 bp) promoter-containing sites 1, 2 and 3, respectively. Antibodies against Histone H3 (D2B12) (#4620) and human RPL30 Exon 3 Primers (#7014) were purchased from from Cell Signalling (Danvers, USA).

For re-ChIP, the beads were eluted with 20 mM dithiothreitol at 37 °C for 30 min, and the eluates were diluted 30-fold for further incubation with the appropriate secondary antibody (HDAC2 or p300) and beads. Gene enrichment was quantified relative to input controls by qPCR using primers specific for the promoter regions of *miR-500a-5p*. Results are shown as FC of qPCR value over IgG. The primers used are listed in Supplementary Table 1.

**Co-immunoprecipitation (Co-IP).** The lysates of cells without or with a stable transfection of tagged constructs were incubated first with 3 μg antibody for 3 h at 4 °C, followed by the precleared protein A/G-agarose bead (Roche, Mannheim, Germany) slurry. After extensive washing, the samples were subjected to western blot analysis for the detection of potential interacting proteins.

**Cell migration and invasion analysis.** Cell migration and invasion assays were also performed. Briefly, transfected cells were seeded at $1 × 10^6$ per well into six-well plates and grown to confluence. The monolayer was then wounded with a pipette tip, and the cells that became detached upon wounding were carefully rinsed away. Next, the medium was changed to remove cell debris, and the cells were cultured in the presence of 10 μg/ml mitomycin C to inhibit cell proliferation. Photographs were taken and migration index was calculated as follows: migration

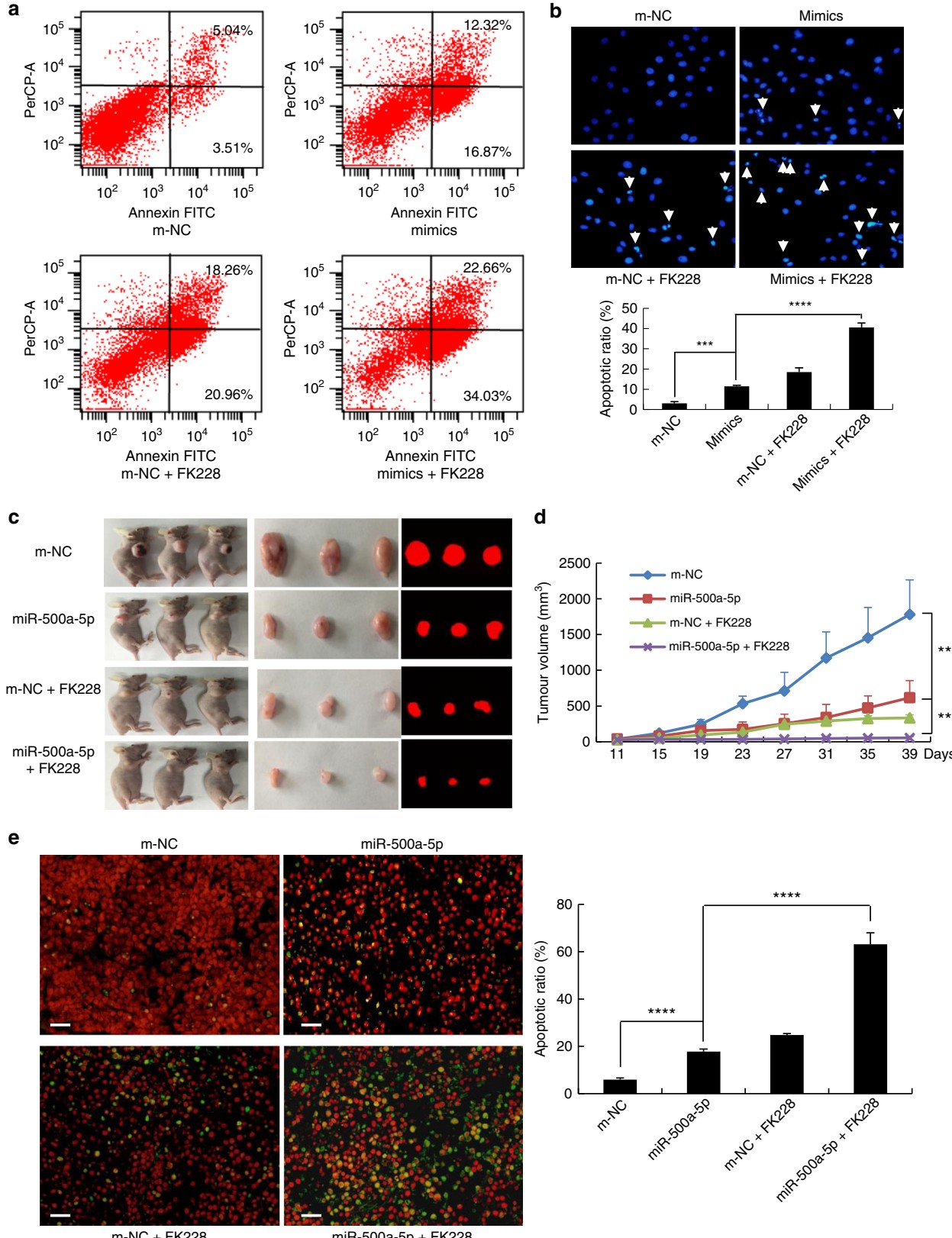

index = [(initial wound width – width of wound at the time point tested)/initial wound width] × 100%.

Cells from the serum-free medium ($1 \times 10^5$ cells/100 µl) were added to the top chamber of each 8-mm-pore transwell chamber (Corning Star; Cambridge, Mass, USA). The bottom chamber was prepared using 10% foetal bovine serum as a chemoattractant. Cells were allowed to migrate through the porous membrane for 20 h at 37 °C. The cells that stuck to the lower surface of the membrane were treated with a fixation/staining solution (0.1% crystal violet, 1% formalin, and 20% ethanol) for visualisation. Cells were quantified as the average number of cells found in five random microscopic high power fields in three independent inserts.

**Flow cytometry analysis**. For cell cycle analysis, CRC cells were reverse transfected in six-well dishes and incubated for 48 h under standard growth conditions.

**Fig. 7** Treatment with miR-500a-5p sensitises cancer cells to FK228-treated apoptosis. **a** LoVo cells were transfected with m-NC or miR-500a-5p for 36 h and then FK228 induced for 48 h, double-stained with Annexin V-FITC and PI, and assessed by flow cytometry analysis to evaluate apoptosis. **b** LoVo cells transfected with m-NC or miR-500a-5p were then FK228 induced for 48 h. Nuclei were stained with Hoechst 33258 and visualised under a fluorescence microscope (the arrow indicates cells with nuclear fragmentation and condensed chromatin). Student's $t$ test; ***$P < 0.01$, m-NC vs miR-500a-5p and ****$P < 0.001$, miR-500a-5p vs miR-500a-5p/FK228. **c** Fluorescence images of subcutaneous tumours of mice injected with LoVo/m-NC cells, LoVo/miR-500a-5p cells, LoVo/ m-NC cells + FK228 and LoVo/miR-500a-5p + FK228 cells. The mice were sacrificed and the photographs shown were taken on day 39. **d** Tumour size was measured starting from the 11th day after tumour cell inoculation in each group. Student's $t$ test; **$P < 0.05$, m-NC vs miR-500a-5p and **$P < 0.05$, miR-500a-5p vs miR-500a-5p/FK228. **e** Apoptosis in tumour tissues as visualised by TUNEL staining (apoptotic cells are indicated in green). The apoptotic index was calculated as the number of apoptotic cells/total number of nucleated cells × 100%. Student's $t$ test; ****$P < 0.001$, m-NC vs miR-500a-5p and ****$P < 0.001$, miR-500a-5p vs miR-500a-5p/FK228. Results of experiments of (**a**), (**b**) and (**e**) were repeated three to four times with the same results. Scale bars, 50 μm in (**b**) and (**e**)

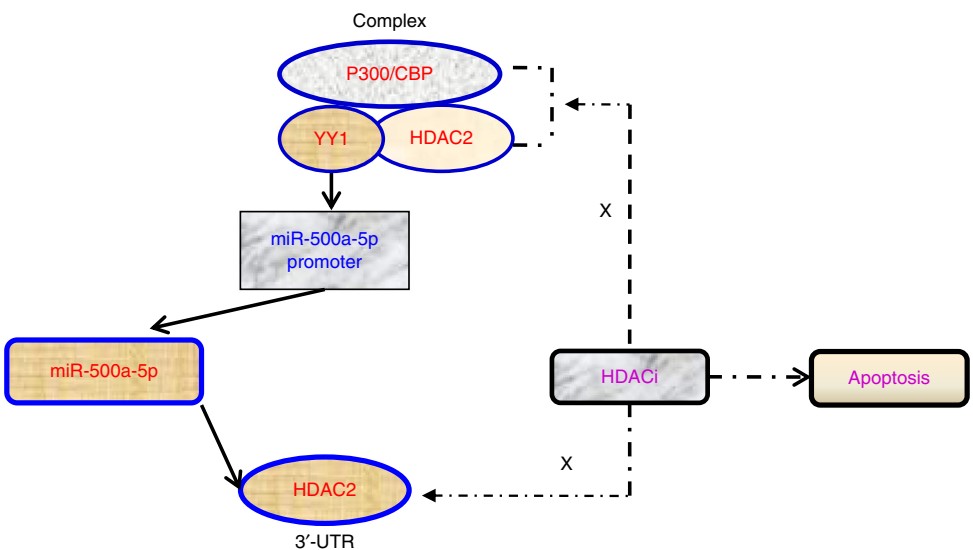

**Fig. 8** Illustration of the hypothesised signal mechanism of miR-500a-5p. Dashed line: HDAC2 was blocked by the HDACi (HDAC inhibitor)

Cells were fixed for 90 min with 70% (v/v) pre-cooled ethanol (−20 °C) and harvested by centrifugation (4 °C, 5 min, 1000 × g). Staining was performed using 69 μM propidium iodide solution in phosphate-buffered saline containing RNaseA (0.6 μg/ml) for 30 min at 37 °C. The cell cycle distribution was calculated from the resultant DNA histograms using FlowJo software.

Apoptosis was detected using the Annexin V-FITC kit (Trevigen Inc., Gaithersburg, MD, USA) according to the manufacturer's instructions. Briefly, cells with various treatments were collected and stained with Annexin V-FITC and PerCP in the dark for 15 min before being subjected to flow cytometry.

**Morphological detection of apoptosis**. Apoptotic cell death was morphologically evaluated as previously described with some modifications[42]. Cells were fixed for 5 min in 3% paraformaldehyde in phosphate-buffered saline. After air drying, the cells were stained for 10 min in Hoechst 33342 (10 μg/ml), mounted in 50% glycerol containing 20 mM citric acid and 50 mM orthophosphate, and stored at −20 °C before analysis. The nuclear morphology was evaluated using a Zeiss IM 35 fluorescence microscope.

**Mass spectrometric analyses**. The lysates of LoVo cells were incubated with 3 g HDAC2 antibody for 3 h at 4 °C followed by incubation with the precleared protein A/G-agarose bead (Roche, Mannheim, Germany) slurry. Proteins samples were visualised following sodium dodecyl sulphate-polyacrylamide gel electrophoresis using a colloidal Coomassie blue stain (Invitrogen) according to the manufacturer's specifications. Selected gel sections were excised, destained in methanol/$H_2O$, and digested in-gel with trypsin (Promega; Madison, WI, USA). Trypsin was added (in 50 mM ammonium bicarbonate) in an approximate ratio 1:25 (enzyme to protein), and in-gel digestion was allowed to continue overnight (37 °C). Peptides were extracted from the gel slices into a 50% acetonitrile solution. Peptides were analysed on a Thermo Scientific Q Exactive mass spectrometer (Fitgene Biotechnology Co., Ltd, Guangzhou, China). The peptides were subjected to nanoelectrospray ionisation followed by tandem mass spectrometry (MS/MS) in a QEXACTIVE (Thermo Fisher Scientific, San Jose, CA, USA) coupled online to the high-performance liquid chromatography. Protein identification was performed with MASCOT software by searching Uniprot_Aedis Aegypti. Peptide mass tolerance values were of 20 ppm and fragment mass tolerances were of 0.6 Da[43].

**Construction and transfection of lentiviral vectors in vivo**. An miR-500a-5p lentiviral expression vector (Ubi-MCS-SV40-Cherry) containing the red fluorescent protein gene (absorption 587 nm, emission 610 nm; GeneChem Co., Ltd., Shanghai, China) was transfected into the lentiviral packaging cell line 293 T. Next, 1 mL viral supernatant containing 5 μg polybrene was added to the CRC cell lines for stable transduction. After 14 days, puromycin-resistant cell pools were established. Ubi-MCS-3FLAG-SV40-EGFP-IRES-puromycin lentiviral expression (absorption 475 nm, emission 505 nm) vector was constructed containing HDAC2 without the 3′-UTR region. Ubi-MCS-3FLAG-SV40-EGFP-IRES-puromycin empty vectors were used as controls. To obtain miR-500a-5p/HDAC2-co-expressing cells, 3 μl of a concentrated HDAC2 lentiviral expression vector solution was added to miR-500a-5p-overexpressing CRC cell lines. Five micrograms per millilitre of polybrene was then mixed with the cells. After 72 h, western blotting was performed to detect HDAC2 expression.

**Tumour growth assay**. All animals were randomly divided into the following five groups: group I, m-NC; group II, miR-500a-5p; group III, vector; group IV, HDAC2; and group V, miR-500a-5p/HDAC2. Lentivirus-transduced LoVo cells were suspended in 100 μL serum-free RPMI and implanted subcutaneously into the flanks of nude mice (three 4- to 6-week-old female BALB/c nu/nu mice in each group; Laboratory Animal Unit, Southern Medical University, China). The fluorescence emitted by cells was collected and imaged using a whole-body RFP/GFP (m-NC-RFP, miR-500a-5p-RFP, vector-GFP, HDAC2-GFP and miR-500a-5p/HDAC2-RFP-GFP) imaging system (Lighttools, Encinitas, CA, USA). The sizes of the resulting tumours were measured weekly. Tumour volumes were calculated as follows: total tumour volume (mm$^3$) = $L \times W^2/2$, where $L$ is the length and $W$ is the width. Thirty-seven days after inoculation, the mice were sacrificed, and their tumours were dissected and weighed. Tissues were microscopically examined by hematoxylin and eosin staining and by immunohistochemistry analysis using anti-Ki-67 and anti-CD105 antibodies.

**TUNEL assay**. Cell apoptosis in tumour xenograft tissues was detected by TUNEL staining (absorption 450–500 nm, emission 515–565 nm)[44]. Tumour tissues from xenografts were excised and immediately formalin-fixed. TUNEL staining was performed using an ApoAlert DNA Fragmentation Assay Kit (Clontech, Palo Alto,

CA, USA) according to the manufacturer's instructions, with apoptotic cells exhibiting green nuclear fluorescence. Tumour tissue sections were counterstained with propidium iodide (PI; absorption 535 nm, emission 615 nm) for 5–10 min to stain cell nuclei. The percentage of apoptotic cells was assessed in 10 randomly selected fields at $40 \times$ magnification[42]. The apoptotic index was calculated as the number of apoptotic cells/total number of nucleated cells $\times 100\%$.

**Statistical analysis**. Statistical analyses were performed using the SPSS statistical software package (standard version 20.0 PSS, Chicago, IL). Comparisons between results from different groups were performed using one-way ANOVA and Dunnett's T3 multiple comparison test. The quantitative data obtained from the experiments with biological replicates are presented as the mean ($\pm$SD). Pearson's $\chi^2$ test was used to analyse the associations of miR-500a-5p expression with clinicopathologic features. HDAC2-miR-500a-5p or YY1-miR-500a-5p interaction tests were performed using linear regression models. Categorical data were analysed using the Fisher's exact test. The survival rates were calculated by the Kaplan–Meier test, and log-rank tests were used to examine the differences in survival rates between the two groups. Two-tailed Student's $t$ test was used to analyse the quantitative data with significant difference being considered if $P$ values were $< 0.05$.

## Data availability
All microarray data are deposited in the Gene Expression Omnibus (GEO database accession no: GSE115108 and GSE122884). All other remaining data are available within the article and supplementary files, or available from the authors upon request.

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

## Acknowledgements
This study was supported by grants from the National Natural Science Funds of China (grant numbers 81672875 & 81772964), high-level topic-matching funds of Nanfang Hospital (grant number 201347) and Guangzhou Pilot Project of Clinical and Translational Research Center (early gastrointestinal cancer, grant number 7415696196402).

## Author contributions
This study was designed and conceived by Jide W., A.L. and S.L. In vitro experiments were performed by W.T., P.Z. and X.W. In vivo experiments were conducted by Jing W., G.L. and J.C. L.X. and Wenjing Z. helped with data analysis. Y.P., X.H. and J.X. were responsible for the collection of specimens. Y.B., L.B., Wei Z., H.G. and C.Y. contributed to technical support. A.L. and S.L. supervised the project and Jide W. wrote the manuscript. Weijie Z. revised the manuscript.

**Additional information**

**Competing interests:** The authors declare no competing interests.

