## [Peer Review File · Nature Communications]

Reviewers' comments:

Reviewer #1 (Remarks to the Author):

The authors of the manuscript "The p300/YY1/miR-500a-5p/HDAC2 signalling axis regulates cell proliferation in human colorectal cancer" describe the role and regulation of miR-500a-5p in colorectal cancer. They found the microRNA to be downregulated in colorectal cancer. As a microRNA target they identified HDAC2 that is frequently upregulated in colon cancer patients and was shown to play an important role in the disease. Both, miR-500a-5p and its target HDAC2 are shown to be able to regulate invasion, migration in vitro and tumor growth in vivo in mice. Furthermore downregulated miR-500a-5p is correlating with clinicopathological parameters like differentiation, lymph node metastasis and TNM staging, linking its downregulation to malignant progression. The authors identified YY1 to be the transcriptional repressor of miR-500a-5p.

There are several points that have to be addressed by the authors.

Major points:

- 1) The authors are talking about possible therapeutic opportunities with the identification of this tumor suppressor pathway. But the authors should first demonstrate that the downregulation of miR-500a-5p is of prognostic relevance. Their follow-up is too short, but I would suggest that they should analyze other datasets like tcga to address this question. So please provide data for OS, PFS.
- 2) The description of the microarray data (microRNA and mRNA) is completely missing in the Methods part. Furthermore, I would suggest including the replicates in the heat maps in the figures.
- 3) ChIP experiments: The authors should absolutely include a region that is not expected to bind as a control to see the background. Also a known positive control would be nice to include. Please show a qPCR analysis for the ChIP experiments to see a kind of quantification of the binding. Figure 6E and F: it is not described which binding site the authors are using in this experiment. Do they see the same effects in all 3 YY1 binding sites?
- 4) Co-IP vs ChIP: the authors are showing a complex between YY1/p300/HDAC2 in a Co-IP, but it was not demonstrated if they are also in a complex when they are bound to DNA. The authors should address this question by at least performing ChIP experiments also with HDAC2 and p300 on the miR-500-5p promoter and maybe even Re-ChIP experiments. Is the binding of HDAC2 for instance changing upon p300 overexpression?
- 5) Is it possible to show the triple complex using endogenous p300?
- 6) Lane 220-222: please revise.
- 7) Lane 244-254: please revise; this part is really confusing

- 8) Figure 1G: here in the FACS analysis there is no increase in sub-G1 detectable; the authors describe only an increase in G1 phase. In Figure 7 instead the authors are analyzing the effect of miR-500a-5p on the apoptosis. Please explain the discrepancy.
- 9) The authors are describing mir-500a-5p to be downregulated in breast. Please check the recent literature.
- 10) To render the anti-correlation between mir-500a-5p and HDAC2 more significant, please perform qPCR/Western blot (or all in qPCR) with all patient samples.
- 11) Figure 6: please show also the WB of the immunoprecipitated p300 in A and HDAC2 in B;
- 12) IHC description is missing
- 13) The genomic location of miR-500a-5p is in a big cluster of microRNAs nearby; are they co-regulated? Is the host gene CLCN5 co-regulated in its expression? Please check.
- 14) Please describe in more detail the plasmid you used for the promoter-luc assay with the YY1 binding sites. Does it contain already a minimal promoter? If so, you can't claim that the region you tested is actually the promoter of miR-500a-5p. In fact, there is for instance one report showing that IL4 upregulated the miR-500a-5p host gene CLCN5 and together with the host gene several microRNAs located together in the third intron ("IL-4 Up-Regulates MiR-21 and the MiRNAs Hosted in the CLCN5 Gene in Chronic Lymphocytic Leukemia"). So you should really check this.
- 15) Co-transfection of YY1 leads to a downregulation of miR-500a-5p. Please check not only the mature form of miR-500a-5p but also the precursors, pri- and pre-miR to show that the downregulation is really transcriptionally.
- 16) The authors should provide a link for the submission of the results of their microarray data to GEO.

Minor points:

- 1) Lane 78: leave out "all"
- 2) Fig. 3A: One Mutant should be Mutant 2 (writing error)
- 3) Lane 197: change "scanned software"; write both software that you used for the transcription factor putative binding site analysis.
- 4) Lane 212: Figure 5E, not 6E
- 5) Figure 5H: the letter H is missing in the Figure
- 6) Lane 267-273: please revise.
- 7) Lane 283: the percentage of apoptotic cells instead of "apoptotic indices".
- 8) Lane 309: "in CRC tissues and " add: ectopic miR-500a-5p expression

- 9) Lane 310-312: please revise; the part where the authors are describing the influence of the triple complex on the miR-500a-5p promoter or miR-500a-5p on the triple complex YY1/p300/HDAC2 is really confusing and should be strongly revised.
- 10) Suppl. Figure 1G: red bars = other cell type? Please check.
- 11) Lane 364-372: please revise.
- 12) Please add in the Discussion also the Figure you are turning back to in your discussion.
- 13) Please add a Figure legend or headline for Figure 8
- 14) Figure 4A: please explain in more detail in the Figure legend
- 15) Figure 4B legend: tumor size was probably not measured after 4 days (see time scale)
- 16) Figure 4 legend: how many μm is the scale bar in C?
- 17) Figure 6 headline: Please delete "restoration of"
- 18) Figure 7C: explain in more detail
- 19) Figure 7D: "four days", see comment 16).
- 20) Lane 494: ATCC, not ATCCC
- 21) Method part reagents: please add the antibodies for HDAC2 and HA.
- 22) Method part qPCR: please add the normalization procedure for YY1 expression and add the citation for Suppl. Table 1.
- 23) Lane 568: in all five analyses (not 4)
- 24) Please add where the mimics/inhibitors were purchase from.
- 25) Lane 610: please correct 104
- 26) Lane 620: antibodies were probably not used in the luc assay, please correct.

Reviewer #2 (Remarks to the Author):

In the study "The p300/YY1/miR-500a-5p/HDAC2 signalling axis regulates cell proliferation in human colorectal cancer", Wang and colleagues report on the tumour suppressive role of miR-500a-5p in colorectal cancer. The authors demonstrate that miR-500a-5p is downregulated in CRC and subsequently perform a comprehensive analysis of the regulatory network around miR-500a-5p, involving HDAC2, YY1 and p300. They identify HDAC2 as a direct miR-500a-5p target gene, which is responsible for the observed tumour suppressive effects of the miRNA.

Overall, the presented data appear solid, although most of the mechanistic studies were performed in two CRC cell lines. Apart from my doubts about miR-500a-5p delivery as a therapeutic strategy, the authors provide a lot of data that support the role of miR-500a-5p as a tumour suppressor in CRC. However, I have several concerns that need to be addressed.

Major comments:

- In many cases, experimental details are lacking. For instance, in the majority of the figures it is not clear how many times the experiments were performed. In addition, information about the type of statistical tests that were used to calculate the presented p-values are frequently lacking.
- The authors mainly perform their functional experiments in the LoVo CRC cell line and in some cases additionally in SW620 CRC cells. First, all miRNA mimetic experiments should be performed in both cell lines. Second, it is rather remarkable that the authors also perform the miR-500a-5p inhibition experiments in these cell lines, whereas they demonstrate that these cell lines have very low levels of this miRNA. All miRNA inhibition experiments should therefore be performed in cells that express miR-500a-5p at "normal levels". Moreover, the authors should inhibit miR-500a-5p expression in non-transformed colon epithelial cells (e.g. organoids), which provides a much cleaner model system than the overexpression experiments in cancer cell lines presented in the manuscript. What is the effect of miR-500a-5p downregulation in healthy colon epithelium? Similarly, is this effect recapitulated by HDAC2 overexpression?
- MiRNAs are well-known for fine-tuning transcript levels, where each individual miRNA targets many different transcripts. This is also confirmed in Fig. 2B, where the authors present a list of transcripts downregulated upon miR-500a-5p expression. Can the observed tumour suppressive effect of miR-500a-5p be fully attributed to HDAC2? Did the authors follow up on any of the other target genes?
- The authors demonstrate that there is a correlation between clinicopathological parameters of colorectal cancer and miR-500a-5p expression. If the entire p300/YY1/miR-500a-5p/HDAC2 signalling axis is involved, there should also be a correlation between elevated HDAC2 and YY1 expression and the same clinicopathological parameters. These analyses should be included.
- HDACs regulate transcription of a large spectrum of genes via histone deacetylation. However, based on the IHC images in Fig. 4E it seems that HDAC2 expression is mainly cytoplasmic, possibly suggesting that its target(s) reside in the cytoplasm and HDAC2 does not exert its effect via histone deacetylation. In Fig. 6A, the authors demonstrate that they can efficiently pull-down HDAC2 binding partners via immunoprecipitation. The authors should screen for other HDAC2 interaction

partners by, for instance, mass spectrometry, as this could provide more specific therapeutics targets.

Minor comments:

- The authors state that miR-500a-5p expression was localized to the cytoplasm of CRC cells. However, this is not clear from the provided images in Fig. 1E: in. Clearer images should be provided with higher magnification. In addition, the authors should include CRC tissue which, from their qRT-PCRs, did not show downregulated miR-500a-5p expression.
- The last sentence of the first paragraph of the Results section "The above findings in CRC" should be rephrased. Up- and downregulation of many genes can be observed in tumours compared to healthy tissue, but this does not say anything about a potential oncogenic or tumour suppressive role.
- Sentence 163-164: "Finally, ... CRC". I assume the authors mean miR-500a-5p mediated HDAC2 downregulation results in proliferation arrest?
- It is not clear what the negative controls for miRNA inhibition or miRNA mimetics (m-NC, i-NC) are. In any case, as negative control for miR-500a-5p mimetic a miR-500a-5p in which the seed sequence is mutated should be used.
- Full scans of the western blots should be provided.

Responses to reviewers' comments:

Reviewer #1 (Remarks to the Author):

The authors of the manuscript “The p300/YY1/miR-500a-5p/HDAC2 signalling axis regulates cell proliferation in human colorectal cancer” describe the role and regulation of miR-500a-5p in colorectal cancer. They found the microRNA to be downregulated in colorectal cancer. As a microRNA target they identified HDAC2 that is frequently upregulated in colon cancer patients and was shown to play an important role in the disease. Both, miR-500a-5p and its target HDAC2 are shown to be able to regulate invasion, migration in vitro and tumor growth in vivo in mice. Furthermore downregulated miR-500a-5p is correlating with clinicopathological parameters like differentiation, lymph node metastasis and TNM staging, linking its downregulation to malignant progression. The authors identified YY1 to be the transcriptional repressor of miR-500a-5p. There are several points that have to be addressed by the authors.

Major points:

1) The authors are talking about possible therapeutic opportunities with the identification of this tumor suppressor pathway. But the authors should first demonstrate that the downregulation of miR-500a-5p is of prognostic relevance. Their follow-up is too short, but I would suggest that they should analyze other datasets like tcga to address this question. So please provide data for OS, PFS.

Answer: We have added data for OS and PFS in the text and colored in red. Thanks for reviewer's advice.

2) The description of the microarray data (microRNA and mRNA) is completely missing in the Methods part. Furthermore, I would suggest including the replicates in the heat maps in the figures.

Answer: We appreciate the reviewer's comments and suggestions. We have made necessary revision in the section of “Method” and colored it in red. Actually, all of our experiments with three replicates were performed in intestinal cell. We have made necessary revision to plot as heat-maps in fig 1a and fig 2b.

3) ChIP experiments: The authors should absolutely include a region that is not expected to bind as a control to see the background. Also a known positive control would be nice to include. Please show a qPCR analysis for the ChIP experiments to see a kind of quantification of the binding. Figure 6E and F: it is not described which binding site the authors are using in this experiment. Do they see the same effects in all 3 YY1 binding sites?

Answer: We agree with the reviewer. We redid the ChIP-qPCR experiments with anti-YY1 antibody. The PCR primers located in exon 3 of RPL30 and anti-Histone H3 antibody served as positive control. Primers were used to amplify the served region containing the distant upstream miR-500a -5p promoter was as the background. The miR-500a-5p promoter region in all 3 YY1 binding sites exhibited significant enrichment after immunoprecipitation with an anti-YY1 antibody. No bands were evident in immunoprecipitates obtained control IgG (Figure 5d& Supplementary Figure 8c). ChIP-qPCR result that is representative of three similar replicates in the revised manuscript.

Moreover, we have added with binding site1 into the Figure 6g and h (previously Figure e and f) and colored in red. In addition, we demonstrate that YY1 protein associates with the miR-500a-5p promoters region in all three YY1 binding sites in CRC cells as shown in Figure 5d and Supplementary Figure 8c. We apologize for the misunderstanding.

4) Co-IP vs ChIP: the authors are showing a complex between YY1/p300/HDAC2 in a Co-IP, but it was not demonstrated if they are also in a complex when they are bound to DNA. The authors should address this question by at least performing ChIP experiments also with HDAC2 and p300 on the miR-500-5p promoter and maybe even Re-ChIP experiments? . Is the binding of HDAC2 for instance changing upon p300 overexpression?

Answer: Thank you for pointing out this. In the revised version, we redid the ChIP-qPCR experiments with anti-HDAC2 and anti-p300 antibody. We demonstrate that YY1 binding was associated with HDAC2 and p300 to miR-500a-5p promoter region in all three YY1 binding sites in both LoVo and SW620 cells using a re-ChIP assay (Supplementary Figure 9 a &b).

In addition, ectopic expression of p300 induced less associated HDAC2 to the miR-500a-5p-site 1 promoter in CRC cells (Supplementary Figure 10c&d). We have added it into the text and Supplementary Figure Legends in Supplementary Figure 9 & 10 in colored in red.

5) Is it possible to show the triple complex using endogenous p300?

Answer: We have added in the text by stating “To further establish endogenous p300 and HDAC2 interactions, we immunoprecipitated cell lysates from CRC cells with HDAC2- specific or p300-specific antibodies. We found endogenous p300 can associate with HDAC2 along with YY1 in CRC cells in a reciprocal manner (Figure 6c & d)”. We appreciate the reviewer’s suggestions.

6) Lane 220-222: please revise.

Answer: We have corrected this mistake in colored in red. Thanks.

7) Lane 244-254: please revise; this part is really confusing

Answer: We apologize for the misunderstanding. We have made necessary revision in the text by stating “The results revealed that over-expression of p300 promoted,

whereas YY1 or HDAC2 inhibited the expression of miR-500a-5p. Moreover, ectopic expression of p300 in combination with YY1 or/and HDAC2 could partially reversed the expression of miR-500a-5p (Figure 6e)” and colored it in red.

8) Figure 1G: here in the FACS analysis there is no increase in sub-G1 detectable; the authors describe only an increase in G1 phase. In Figure 7 instead the authors are analyzing the effect of miR-500a-5p on the apoptosis. Please explain the discrepancy.

Answer: Thank you. Our manuscript showed that cell cycle progression was performed by FACS analysis as Reviewers' comments Figure 1a. FACS gate was set on population after excluding apoptotic cells and cell debris as Reviewers' comments Figure 1b.

Reviewers' comments

Figure 1

In the revised manuscript, cell cycle progression was performed using a FACS analysis. We used a gate on the cell population comprising of apoptotic cells, apoptotic bodies and debris excluding unwanted cell and the cell population are in sub-G 0/1 region. We have corrected Reviewers' comments Figure 2 and Supplementary Figure 3a. All of these experiments were repeated 3 times with identical findings.

9) The authors are describing mir-500a-5p to be downregulated in breast. Please check the recent literature.

Answer: We checked the recent literature on this topic of mir-500a-5p in breast cancer (Degli Esposti D, et al. miR-500a-5p regulates oxidative stress response genes in breast cancer and predicts cancer survival. *Sci Rep.* 2017; 7(1):15966).

MiR-500a-5p and miR-500a-3p expression levels are quite heterogeneous across the

different cell lines in all cell lines. MCF-7 cells display the highest level of expression of both strands, while T47D cells showed the lowest. Therefore, we deleted a sentence that “including down-regulation in breast cancer”. We have revised the text by stating that “Studies have reported miR-500a-5p dysregulation in several cancers, including liver cancer and gastric cancer” and colored it in red. We apologize for the misunderstanding.

10) To render the anti-correlation between mir-500a-5p and HDAC2 more significant, please perform qPCR/Western blot (or all in qPCR) with all patient samples.

Answer: In the revised manuscript, we performed q-RT-PCR experiments with all patient samples and have revised the text by stating “the mRNA level of HDAC2 in the CRC samples obtained from 81 patients was negatively correlated to the miR-500a-5p expression level (Figure 2e)” and “YY1 protein expression was negatively correlated with miR-500a-5p expression (Figure 5f)” and colored it in red. Thanks for reviewer’s advice.

11) Figure 6: please show also the WB of the immunoprecipitated p300 in A and HDAC2 in B;

Answer: We have performed experiments using the co-immunoprecipitated assays and have added this figure 6 into Panel a & b. Thanks.

12) IHC description is missing

Answer: We have added in Fig 5h. Thanks a lot.

13) The genomic location of miR-500a-5p is in a big cluster of microRNAs nearby; are they co-regulated? Is the host gene CLCN5 co-regulated in its expression? Please check.

Answer: We agree with the reviewer. In the revised manuscript, we performed a q-RT-PCR assay and assessed whether expression of miR-500a-5p was associated with miR-362, miR-502, miR-532 or CLCN5 expression (“IL-4 Up-Regulates MiR-21 and the MiRNAs Hosted in the CLCN5 Gene in Chronic Lymphocytic Leukemia”) in colon epithelial cell lines NCM460, FHC, SW480, DLD-1, SW1116, SW620, HCT1116, LoVo and Caco2 cells. We showed that the level of miR-500a-5p positive correlated to the miR-502 (Reviewers' comments Figure 3a, $r = 0.754$, $p = 0.019$) expression level. Surprisingly, the expression of miR-500a-5p was not correlated with miR-532 (Reviewers' comments Figure 3b, $r = 0.574$, $p = 0.106$), miR-362 (Reviewers' comments Figure 3c, $r = 0.661$, $p = 0.052$) and CLCN5 (Reviewers' comments Figure 4, $r = -0.272$, $p = 0.419$). We have added it into the text and colored in red.

Reviewers' comments Figure 3

Reviewers' comments Figure 4

14) Please describe in more detail the plasmid you used for the promoter-luc assay with the YY1 binding sites.

Answer: We have made necessary revision in the section of “Materials and methods” and have added to Supplementary Table 1 in colored in red. Thanks for your carefulness.

15) Does it contain already a minimal promoter? If so, you can't claim that the region you tested is actually the promoter of miR-500a-5p. In fact, there is for instance one report showing that IL4 upregulated the miR-500a-5p host gene CLCN5 and together with the host gene several microRNAs located together in the third intron (“IL-4 Up-Regulates MiR-21 and the MiRNAs Hosted in the CLCN5 Gene in Chronic Lymphocytic Leukemia”). So you should really check this.

Answer: The reviewer is very professional. In the revised version, we have done extra cell q-RT-PCR and promoter experiments. Until some years ago, intronic miRNAs

were generally thought to be processed from the host-gene transcript, with the intronic miRNA and its host gene showing concordant expression levels, since driven by the same promoter. However, subsequent studies showed many examples of poor correlation in expression levels between miRNAs and their host genes, a phenomenon that could be easily explained by the presence of specific promoters driving the expression of intronic miRNAs. MiR-500a-5p genes located within the third intron of the CLCN5 gene. We evaluated that the expression relationships between miR-500a-5p and CLCN5 in 20 CRC tissues. We found that the expression of miR-500a-5p was no relevant to CLCN5 ($r = -0.405$, $p = 0.077$) (Supplementary Figure 7). Therefore, we analysed the 1-kb range of genomic DNA upstream of the miR-500a-5p in the promoter region (<http://genome.ucsc.edu/>). This experiment showed that the miR-500a-5p promoter is able to drive a significantly higher luciferase expression (about 18.2 in loVo cells and 15.7 in SW620 cells folds) than the empty vector (pGL3-basic), used as control (Supplementary Figure 8a, $P < 0.001$). Thus, miR-500a-5p expression is driven by its own promoter and colored in red. Thanks a lot.

16) Co-transfection of YY1 leads to a downregulation of miR-500a-5p. Please check not only the mature form of miR-500a-5p but also the precursors, pri- and pre-miR to show that the downregulation is really transcriptionally.

Answer: This is a good point. As suggested, we did extra experiments by q-RT-PCR assay. We showed that transient expression of YY1 led to decreased expression only the mature form (Figure 5g) but also precursors or primary of miR-500a-5p in LoVo and SW620 cells (Supplementary Figure 8d). We have added the oligonucleotides primers in the section of “Supplementary Table 1” and colored it in red.

17) The authors should provide a link for the submission of the results of their microarray data to GEO.

Answer: All microarray data are deposited in the Gene Expression Omnibus (GEO data base accession No: GSE115108). We appreciate the reviewer’s suggestions.

Minor points:

1) Lane 78: leave out “all”

Answer: we deleted the word “all”. Thanks a lot.

2) Fig. 3A: One Mutant should be Mutant 2 (writing error)

Answer: We have corrected this mistake. Thanks.

3) Lane 197: change “scanned software”; write both software that you used for the transcription factor putative binding site analysis.

Answer: We have corrected this mistake. Thanks.

4) Lane 212: Figure 5E, not 6E

Answer: We have corrected this mistake. Thanks.

5) Figure 5H: the letter H is missing in the Figure

Answer: We have corrected this mistake. Thanks.

6) Lane 267-273: please revise.

Answer: Thanks for reviewer's advice and has corrected this minor error in the text and colored in red.

7) Lane 283: the percentage of apoptotic cells instead of "apoptotic indices".

Answer: We have corrected this mistake. Thanks.

8) Lane 309: "in CRC tissues and " add: ectopic miR-500a-5p expression

Answer: We have corrected this mistake. Thanks.

9) Lane 310-312: please revise; the part where the authors are describing the influence of the triple complex on the miR-500a-5p promoter or miR-500a-5p on the triple complex YY1/p300/HDAC2 is really confusing and should be strongly revised.

Answer: We have corrected this mistake in colored in red. Thanks a lot.

10) Suppl. Figure 1G: red bars = other cell type? Please check.

Answer: Suppl. Figure 1g: red bars was LoVo cells (Supplementary Figure 4b). Thanks for your carefulness.

11) Lane 364-372: please revise.

Answer: We appreciate the reviewer's suggestion. We have carefully revised in the section of "Discussion" and colored it in red. Thanks

12) Please add in the Discussion also the Figure you are turning back to in your discussion.

Answer: It is a very good suggestion. We try to follow the advice and have revised using red colored text. Thanks a lot.

13) Please add a Figure legend or headline for Figure 8.

Answer: We have added to headline for Figure 8 and a Figure legend in colored in red. Thanks for your carefulness.

14) Figure 4A: please explain in more detail in the Figure legend

Answer: We have added the legends in Figure 4a and colored it in red. Thanks a lot.

15) Figure 4B legend: tumor size was probably not measured after 4 days (see time scale)

Answer: We have added the legends for each color in Figure 4b. Thanks a lot.

16) Figure 4 legend: how many μm is the scale bar in C?

Answer: Thanks. We insert a scale bar in Figure 4c and colored it in red.

17) Figure 6 headline: Please delete "restoration of"

Answer: Thanks for your carefulness. We deleted "restoration of".

18) Figure 7C: explain in more detail

Answer: We have added in Fig 7c and colored it in red. Thanks.

19) Figure 7D: "four days", see comment 16).

Answer: We have added the legends for each color in Figure 7d. Thanks a lot.

20) Lane 494: ATCC, not ATCCC

Answer: We have corrected this mistake. Thanks a lot.

21) Method part reagents: please add the antibodies for HDAC2 and HA.

Answer: We have added the antibodies for HDAC2 and HA. Thanks a lot.

22) Method part qPCR: please add the normalization procedure for YY1 expression and add the citation for Suppl. Table 1.

Answer: We have added in Suppl. Table 1 and some valuable references.

23) Lane 568: in all five analyses (not 4)

Answer: We have corrected this mistake. Thanks.

24) Please add where the mimics/inhibitors were purchase from.

Answer: We have added the Company in the section of "Materials and Methods" and colored it in red. Thanks.

25) Lane 610: please correct 104

Answer: We have corrected this mistake. Thanks.

26) Lane 620: antibodies were probably not used in the luc assay, please correct.

Answer: We have corrected this mistake. Thanks.

Reviewer #2 (Remarks to the Author):

In the study "The p300/YY1/miR-500a-5p/HDAC2 signalling axis regulates cell proliferation in human colorectal cancer", Wang and colleagues report on the tumour suppressive role of miR-500a-5p in colorectal cancer. The authors demonstrate that miR-500a-5p is downregulated in CRC and subsequently perform a comprehensive analysis of the regulatory network around miR-500a-5p, involving HDAC2, YY1 and p300. They identify HDAC2 as a direct miR-500a-5p target gene, which is responsible for the

observed tumour suppressive effects of the miRNA. p as a tumour suppressor in CRC. However, I have several concerns that need to be addressed.

Major comments:

- In many cases, experimental details are lacking. For instance, in the majority of the figures it is not clear how many times the experiments were performed. In addition, information about the type of statistical tests that were used to calculate the presented p-values are frequently lacking.

Answer: Actually, all of our experiments were performed at least three times with the same results. We have added it into the Figure Legends.

Moreover, we have added in the related “Materials and methods” text using statistical tests colored in red. We apologize for the misunderstanding. We have performed a statistical analysis in figure. Thanks a lot for the advice.

- The authors mainly perform their functional experiments in the LoVo CRC cell line and in some cases additionally in SW620 CRC cells. First, all miRNA mimetic experiments should be performed in both cell lines.

Answer: Thank you for pointing out this. In the revised manuscript, we performed miR-500a-5p mimetic experiments including colony-forming, EdU, wound healing and invasion assays both LoVo and SW620 cells. We have made revision shown in Fig 1f & h, Supplementary Figure 2a, c & e and Supplementary Figure 4a & b. We made necessary interpretation in RESULT section and Figure Legends in colored in red.

Second, it is rather remarkable that the authors also perform the miR-500a-5p inhibition experiments in these cell lines, whereas they demonstrate that these cell lines have very low levels of this miRNA. All miRNA inhibition experiments should therefore be performed in cells that express miR-500a-5p at "normal levels". Moreover, the authors should inhibit miR-500a-5p expression in non-transformed colon epithelial cells (e.g. organoids), which provides a much cleaner model system than the overexpression experiments in cancer cell lines presented in the manuscript. What is the effect of miR-500a-5p downregulation in healthy colon epithelium?

Answer: The reviewer is very professional. We replaced human colorectal cancer cells LoVo and SW620 with non-transformed colon epithelial cells NCM460 and FHC. A series of functional experiments performed using miR-500a-5p inhibition assay in NCM460 and FHC. We have made revision shown in Fig 1g & i, Supplementary Figure 2b, d & f and Supplementary Figure 4c & d. We made necessary interpretation in RESULT section and Figure Legends in colored in red.

Similarly, is this effect recapitulated by HDAC2 overexpression?

Answer: Thank you. We have revised the text and confirmed that HDAC2 is a target gene of miR-500a-5p. The results revealed that expression of this protein was down-regulated in the CRC cells transfected with miR-500a-5p mimics compared with the m-NC cells, but that it was inversely up-regulated in the normal human colon

epithelial cells FHC and NCM460 transfected with the miR-500a-5p inhibitor (Figure 3c). In addition, the FHC and NCM460 cell in those transfected with the miR-500a-5p inhibitor was significantly increased, whereas HDAC2 knockdown in miR-500a-5p inhibitor cells decreased the proliferation and invasion of miR-500a-5p inhibitor cells (Supplementary Figure 5a & b). These data suggest that miR-500a-5p might inhibit HDAC2 protein expression through the 3'-UTR at the posttranscriptional level and colored in red.

- MiRNAs are well-known for fine-tuning transcript levels, where each individual miRNA targets many different transcripts. This is also confirmed in Fig. 2B, where the authors present a list of transcripts downregulated upon miR-500a-5p expression. Can the observed tumour suppressive effect of miR-500a-5p be fully attributed to HDAC2? Did the authors follow up on any of the other target genes?

Answer: To determine whether tumour suppressive effect of miR-500a-5p be fully attributed to HDAC2, we follow up on any of the other target genes such as XIAP and RICTOR. Our western blotting data demonstrated that the expression of XIAP or RICTOR is decreased in all tested cell types treated with miR-500a-5p (Figure 6). Our results suggested that expression of miR-500a-5p be partly attributed to HDAC2, XIAP or RICTOR. Thank you.

- The authors demonstrate that there is a correlation between clinicopathological parameters of colorectal cancer and miR-500a-5p expression. If the entire p300/YY1/miR-500a-5p/HDAC2 signalling axis is involved, there should also be a correlation between elevated HDAC2 and YY1 expression and the same clinicopathological parameters. These analyses should be included.

Answer: We appreciate the reviewer's suggestion. We evaluated the relationship between HDAC2 (Supplementary Table 3) or YY1 (Supplementary Table 4) expression and the same clinicopathological parameters. We have revised the text and colored in red. Thanks.

- HDACs regulate transcription of a large spectrum of genes via histone deacetylation. However, based on the IHC images in Fig. 4E it seems that HDAC2 expression is mainly cytoplasmic, possibly suggesting that its target(s) reside in the cytoplasm and HDAC2 does not exerts its effect via histone deacetylation.

Answer: We examined the expression of Ki-67 and CD105 protein by immunohistochemistry in the xenograft tumors. We are not HDAC2 protein expression. Representative images of the tumours after IHC staining are shown in Figure 4c (Ki-67) and e (CD105). Thank you for pointing out this.

- In Fig. 6A, the authors demonstrate that they can efficiently pull-down HDAC2 binding partners via immunoprecipitation. The authors should screen for other HDAC2 interaction partners by, for instance, mass spectrometry, as this could provide more specific therapeutics targets.

Answer: It is a practical suggestion. We performed LC-MS/MS analysis in LoVo

cells. The results indicated that 190 proteins precipitated with HDAC2 antibody were detected when compared to those precipitated with the irrespective IgG. A detailed summary of these proteins were given in Supplementary Table 5. We have made necessary revision in the section of “Results” and colored it in red. We have added some valuable references.

Minor comments:

1.- The authors state that miR-500a-5p expression was localized to the cytoplasm of CRC cells. However, this is not clear from the provided images in Fig. 1E: in. Clearer images should be provided with higher magnification.

Answer: The reviewer is very professional. We have shown the images with higher magnification and miR-500a-5p was localized in both nuclei and cytoplasm of CRC cells. Thank you.

2. In addition, the authors should include CRC tissue which, from their qRT-PCRs, did not show downregulated miR-500a-5p expression.

Answer: We have revised the text by stating “The results revealed that expression of this miRNA was down-regulated by up to 7.67-fold in 64 of the 81 CRC samples by qRT-PCR (Figure 1c). Its expression was significantly lower in CRC patient tissues compared with the adjacent normal colon mucosa tissues (Figure 1d).” and colored it in red. Thank you.

3.- The last sentence of the first paragraph of the Results section "The above findings in CRC" should be rephrased. Up- and downregulation of many genes can be observed in tumours compared to healthy tissue, but this does not say anything about a potential oncogenic or tumour suppressive role.

Answer: We have corrected this mistake. Thank you.

4.- Sentence 163-164: "Finally, ... CRC". I assume the authors mean miR-500a-5p mediated HDAC2 downregulation results in proliferation arrest?

Answer: We have revised the text by stating "Finally, we determined that whether miR-500a-5p-mediated HDAC2 regulated proliferation and metastasis in CRC cells". Thank you.

5. - It is not clear what the negative controls for miRNA inhibition or miRNA mimetics (m-NC, i-NC) are. In any case, as negative control for miR-500a-5p mimetic a miR-500a-5p in which the seed sequence is mutated should be used.

Answer: The reviewer is very professional. We have added the sequence in the section of “Materials and Methods” and colored it in red.

6. - Full scans of the western blots should be provided.

Answer: Thank you. Full scans of the western blots are provided in Supplementary Figure.

** See Nature Research's author and referees' website at www.nature.com/authors for information about policies, services and author benefits

This email has been sent through the Springer Nature Tracking System
NY-610A-NPG&MTS

Confidentiality Statement:

This e-mail is confidential and subject to copyright. Any unauthorised use or disclosure of its contents is prohibited. If you have received this email in error please notify our Manuscript Tracking System Helpdesk team at <http://platformsupport.nature.com> .

Details of the confidentiality and pre-publicity policy may be found here <http://www.nature.com/authors/policies/confidentiality.html>

Privacy Policy | Update Profile

Reviewers' comments:

Reviewer #1 (Remarks to the Author):

The revised version of the manuscript "The p300/YY1/miR-500a-5p/HDAC2 signalling axis regulates cell proliferation in human colorectal cancer" was significantly improved by the authors. Anyway, there are still several points that need to be addressed. In general some parts need definitely a revision of the English language (see below).

Major points:

1) The role of p300 in the regulation of miR-500a-5p is still puzzling. On the one hand it binds via YY1 to the DNA in the miR-500a-5p promoter, proven by the Re-ChIP experiments. But on the other hand, how do the authors explain the fact that p300 overexpression leads to an increase of p300 binding to DNA in the ChIP experiment while YY1 binding is decreased?

2) Why should overexpression of p300 and YY1 decrease miR-500a-5p expression compared to overexpression of p300 alone?

3) Fig. 6g and h: the distant region as control should be included in the ChIP experiment.

4) The precise description of the various luc constructs is still missing. Did the authors use the basic pGL3 for the promoter analysis? And did they use instead a pGL3 vector containing a promoter for the UTR studies? The authors used somehow Renilla luciferase as control but they did not mention if they co-transfected the corresponding vector. Furthermore, the authors describe in 2 paragraphs the luc assays in the Mat/Met part. They should combine them or use different titles distinguishing between UTR/promoter analysis.

Again, the vector construction should be described in more detail. In the Suppl. Table 1 there is only one R primer listed for all luc promoter constructs. How is it possible? And why is this one changed compared to the previous manuscript version without indicating it in red?

Please explain in detail the vector cloning strategy.

5) The authors should describe in the Materials and Methods section how they calculated the relative binding for the ChIP and Re-ChIP experiments. Furthermore, they should describe how they calculated the migration and invasion index (different in Suppl. Fig. 4b and 4d).

6) The description of the IHC method is still missing. The cited article 43 does not contain a description of the IHC method.

7) The authors should describe in the Mat/Met section the colony formation assay. They should also take care that in some figure legends they describe it as "anchorage independent colony formation". So did they use soft agar? In the Results section it is only described as normal colony formation assay.

8) The question if the other 2 binding sites (2 and 3) show a similar behavior like in Fig. 6g and h and in Suppl. Fig. 10 was not answered yet.

9) I would suggest that the authors include all their 81 patients in the analysis of Suppl. Fig. 7.

Minor points:

1) Please add the antibodies Ki-67 and CD105 in the IHC method part.

2) Lane 43, 342, 355 and 437: "FK-228-treated" instead of "FK-228-induced".

3) Lane 70: "and its expression is modulated via the.." instead of "and modulates expression...".

4) Lane 162: "the HDAC2 gene" instead of "HDAC2 genes".

- 5) Please revise the following lanes: 185-186; 194-195; 207-208; 228-229; 251-252; 261-262; 294-298; 310; 330; 332; 333-335; 427; 433-435; 488-489; 722-723; 729-731; 896-897; 926-928.
- 6) Lane 233: "miR-500a-5p gene" instead of miR-500a-5p in the promoter region".
- 7) Lane 289: What do the authors mean with " somatic cells"?
- 8) Lane 348: What is NS?
- 9) Lane 429-430: Please revise.
- 10) Lane 681: "4 attograms (Ag) polybrene" seems very low. Please check.
- 11) Lane 683: The C-terminal construct of HDAC2 is not appearing in the Result section.
- 12) Lane 919: Please mention briefly the microarray; lane 477: Delete " MicroRNA" in the headline as the paragraph describes both types of microarrays, microRNAs and mRNAs.
- 13) Lane 899: Dunnett's T3 multiple comparison test is not mentioned in the Method section.
- 14) Lane 934 and lane 31 in the Suppl. Information: exchange "micrographs" with "results".
- 15) Lane 936 and Suppl. Information lane 33: exchange" inoculation" with " seeding".
- 16) Lane 947: "measured starting from 13 days" instead of "measured at thirteen days".
- 17) Lane 966: "CRC tissues" add: "measured by qPCR".
- 18) Lane 1007: "measured starting from 11 days" instead of "measured 11 days".
- 19) Please revise Lanes 6, 17, 52 and 72 in Suppl. Information.

Reviewer #2 (Remarks to the Author):

The authors have resubmitted a significantly improved manuscript in which they included a substantial amount of new data answering most of my comments. However, a few of my concerns remain.

- Besides HDAC2, the authors now demonstrate that XIAP and RICTOR are miR-500a-5p targets as well. Their concluding sentence (194-195) from this data is confusing. Moreover, since XIAP and RICTOR are miR-500a-5p targets, do XIAP and RICTOR also functionally contribute to the tumour suppressive effects of miR-500a-5p in CRC cells? Or is it just HDAC2 that is involved?
- Following my request, the authors performed mass spectrometry to find additional HDAC2 interaction partners. Remarkably, within the 190 potential interaction partners, YY1 and p300 were not identified. The authors should comment on this.
- Fig. 1E, the miRNA ISH images are still not clear. The more intense "signal" in healthy tissue might as well be caused by differences in counter stain intensity. They should therefore provide images of miRNA ISH done on healthy and neighbouring tumour tissue on the same slide.
- There are two Luciferase assay descriptions in Methods section.
- Line 110: miR-5001-5p should be miR-500a-5p.

Reviewers' comments:

Reviewer #1 (Remarks to the Author):

The revised version of the manuscript "The p300/YY1/miR-500a-5p/HDAC2 signalling axis regulates cell proliferation in human colorectal cancer" was significantly improved by the authors. Anyway, there are still several points that need to be addressed. In general some parts need definitely a revision of the English language (see below).

Major points:

1) The role of p300 in the regulation of miR-500a-5p is still puzzling. On the one hand it binds via YY1 to the DNA in the miR-500a-5p promoter, proven by the Re-ChIP experiments. But on the other hand, how do the authors explain the fact that p300 overexpression leads to an increase of p300 binding to DNA in the ChIP experiment while YY1 binding is decreased?

Answer: We thank the reviewer for pointing out this for us. It was previously reported that p300 is mutated in several forms of cancer suggesting a tumor suppressor role for this protein and provides co-repressor function (Krubasik D, et al. absence of p300 induces cellular phenotypic changes characteristic of epithelial to mesenchyme transition. *Br J Cancer*. 2006; 94(9):1326-32 and Muraoka et al. p300 gene alterations in colorectal and gastric carcinomas. *Oncogene* 1996; 12: 1565-1569). Some studies reported that HDAC2 and YY1 significantly overexpressed in gastrointestinal cancers (Zhu P, et al. Induction of HDAC2 expression upon loss of APC in colorectal tumorigenesis. *Cancer Cell*. 2004; 5(5): 455-63 and Zhang N, et al. microRNA-7 is a novel inhibitor of YY1 contributing to colorectal tumorigenesis. *Oncogene*. 2013; 32(42): 5078-88). Based on these studies, there seemed be a negative correlation between functions of p300 and functions of HDAC2 and YY1 in CRC cells. In our studies, we showed that p300, HDAC2 formed a complex with YY1 to bind to the miR-500a-5p promoter YY1 site. Functionally, YY1 and HDAC2 inhibits miR-500a-5p promoter transcription (Figure 5c and Figure 6g & h), whereas, ectopic of expression of p300 weakened YY1 and HDAC2 to bind to the miR-500a-5p promoter YY1-binding site, and activated the miR-500a-5p promoter transcription in deed in GC cells.

We have revised the text in the section of "Discussion" by stating that "It was previously reported that p300 is mutated in several forms of cancer suggesting a tumor suppressor role for this protein. Some reports have shown that p300 exist in multi-molecular complexes *in vivo* and function as co-activators or co-repressor for a variety of HDAC3 and YY1.^{21,38,39} Consistently, we showed that p300, HDAC2 formed a complex with YY1 to bind to the miR-500a-5p promoter YY1 site. Functionally, YY1 and HDAC2 inhibits miR-500a-5p promoter transcription (Figure 5c and Figure 6g & h), whereas, ectopic of expression of p300 weakened YY1 and HDAC2 to bind to the miR-500a-5p promoter YY1-binding site, and activated the miR-500a-5p promoter transcription in deed in GC cells. Therefore, we believe that

p300/YY1/miR-500a-5p/HDAC2 signalling axis plays important roles cancer development” and colored in red.

2) Why should overexpression of p300 and YY1 decrease miR-500a-5p expression compared to overexpression of p300 alone?

Answer: According to our studies, YY1 inhibits miR-500a-5p promoter transcription (Figure 5c and Figure 6g & h), whereas, ectopic of expression of p300 weakened YY1 to bind to the miR-500a-5p promoter YY1-binding site, and activated the miR-500a-5p promoter transcription, that is why overexpression of p300 and YY1 (activator plus inhibitor) decrease miR-500a-5p expression compared to overexpression of p300 alone (activator alone). Thank you for pointing out this.

3) Fig. 6g and h: the distant region as control should be included in the ChIP experiment.

Answer: This is a good point. We redid the ChIP-qPCR experiments in CRC cells expressing exogenous YY1. Primers were used to amplify the sequence region containing the distant upstream miR-500a -5p promoter was as control. The miR-500a-5p promoter region in all 3 YY1 binding sites exhibited significant enrichment after immunoprecipitation with an anti-YY1 antibody. We have corrected Figure 6g & h. Thanks.

4) The precise description of the various luc constructs is still missing. Did the authors use the basic pGL3 for the promoter analysis? And did they use instead a pGL3 vector containing a promoter for the UTR studies? The authors used somehow Renilla luciferase as control but they did not mention if they co-transfected the corresponding vector. Furthermore, the authors describe in 2 paragraphs the luc assays in the Mat/Met part. They should combine them or use different titles distinguishing between UTR/promoter analysis.

Answer: We apologize for the misunderstanding. We use different titles distinguishing as “Luciferase activity assay for the 3’UTR study” and “Promoter Analysis” in the section of “Materials and methods”. We have revised the text by stating that “**Luciferase activity assay for the 3’UTR study:** The luciferase reporter plasmid carrying the wild-type (WT) or mutated (MUT) HDAC2 3’-untranslated region (UTR) (pMIR-report-HDAC2-3’-UTR and pMIR-report-HDAC2-MUT1-3’-UTR, pMIR-report-HDAC2-MUT2-3’-UTR, respectively) was transfected into CRC cells along with the miR-500a-5p mimics using Lipofectamine 2000 (Invitrogen). After transfection (36 - 48 hours), the cells were lysed, and luciferase activity was measured with the Dual-Luciferase Reporter Assay system (Promega, Madison, WI). The sequences of HDAC2-WT-3’-UTR and HDAC2-MUT-3’-UTR are shown in Figure 3a” and “**Promoter Analysis:** The 1-kb region directly upstream of miR-500a-5p was predicted using UCSC software. Analysis of YY1 binding sites on the miR-500a-5p promoter was performed using the TF prediction programme Consite (<http://asp.ii.uib.no:8090/cgi-bin/CONSITE/consite>). MiR-500a-5p promoter (miR-500a-5p-p) construct contained the YY1 binding sites 1 (miR-500a-5p-p-site 1: -333 ~ -327), sites 2 (miR-500a-5p-p-site 2:

-628~ -622), sites 3 (miR-500a-5p-p-site 3: -747 ~ -741) or 1kb (p-Luc-1kb, -1003). Dual luciferase assay was performed using the Dual-luciferase Reporter Assay kit (Promega, Madison, WI) as per the manufacturer's instructions. Briefly, 1×10^5 cells were seeded in each well of a 24-well tissue culture plate. The cells were incubated until 70% confluent. Cells in each well were transfected with 0.8 μg of pGL3 basic or p-Luc-1kb plasmid by 1ml Lipofect-AMINE 2000 reagent. The Renilla luciferase reporter pRL-CMV plasmid (Promega) 0.01 μg per well was cotransfected as the internal control. After transfection for 4 h, cells were transferred into normal medium. To examine the effect of YY1 construct, miR-500a-5p-p-site 1, 2 or 3 reporter plasmid was cotransfected with YY1 or vector. Forty-eight hours later, the cells were treated with passive lysis buffer. Luciferase activities were measured with a luminometer (lumatec LB9507, Berthold, Bad Wildbad, Germany). The firefly luciferase activity value was normalized to the renilla activity value. Promoter transcription activity was presented as the fold induction of relative luciferase unit (RLU) compared with basic pGL3 vector control. (The RLU was the value of the firefly luciferase unit divided by the value of the renilla luciferase unit.) The oligonucleotides primers used in the luciferase activity assays are listed in Supplementary Table 1". We have revised the text and colored in red. Thanks you.

Again, the vector construction should be described in more detail. In the Suppl. Table 1 there is only one R primer listed for all luc promoter constructs. How is it possible? And why is this one changed compared to the previous manuscript version without indicating it in red?

Please explain in detail the vector cloning strategy.

Answer: We apologize for the inconvenience. We have made revision and the primer sequence listed in "Suppl. Table 1" in colored in red. Thank you!

In addition, we described the plasmids cloning strategy in detail in the section of "Materials and methods". Thank you!

5) The authors should describe in the Materials and Methods section how they calculated the relative binding for the ChIP and Re-ChIP experiments. Furthermore, they should describe how they calculated the migration and invasion index (different in Suppl. Fig. 4b and 4d).

Answer: It is a practical suggestion. We have revised the text by stating "Gene enrichment was quantified relative to input controls by qPCR using primers specific for the promoter regions of miR-500a-5p. Results are shown as fold change of qPCR value over IgG."

Moreover, we calculated the migration and invasion index and have revised the text by stating "Cell migration assay: Photographs were taken and migration index was calculated as follows: migration index = [(initial wound width-width of wound at time point tested)/initial wound width] X 100%." and "Cell invasion assay: Cells were quantified as the average number of cells found in five random microscopic fields in three independent inserts."

We have corrected this mistake shown in Supplementary Figure 5b & d (Previously known as Suppl. Fig. 4b and 4d), Figure 3d & f and Supplementary Figure 6a&b. In addition, we have shown in “Materials and Methods” and colored in red. Thank you!

6) The description of the IHC method is still missing. The cited article 43 does not contain a description of the IHC method.

Answer: It is really our fault to miss the exact description for immunohistochemistry in the section of Material and Methods. We have revised the text by stating that “IHC analysis was conducted to determine HDAC2, YY1, Ki-67 and CD105 protein expression in CRC as previously described. Briefly, paraffin-embedded tissue blocks were cut into 5-mm sections and transferred to glass slides. The slides were deparaffinized with xylene, rehydrated with ethanol, washed, and subjected to microwave retrieval in a citrate buffer. Sections were then immersed in 3% hydrogen peroxide to block endogenous peroxidase activity and incubated with the first antibodies followed by incubation with the biotin-linked anti-Rabbit IgG (Dako, Copenhagen, Denmark) in combination with the DAB complex. Normal rabbit or mouse IgG (Sigma) was used as the isotype controls”. and colored in red.

Moreover, we replaced the old reference 43rd (Direct regulation of FOXK1 by C-jun promotes proliferation, invasion and metastasis in gastric cancer cells. Cell Death Dis. 2016; 7(11):e2480) with new one (RUFY3 interaction with FOXK1 promotes invasion and metastasis in colorectal cancer. Sci Rep. 2017, 7(1): 3709) and colored it in red. Thanks you.

7) The authors should describe in the Mat/Met section the colony formation assay. They should also take care that in some figure legends they describe it as “anchorage independent colony formation”. So did they use soft agar? In the Results section it is only described as normal colony formation assay.

Answer: The reviewer is very professional and we apologize for the misunderstanding. We have made necessary revision in the section of “Materials and methods” and have revised the text by stating “Colony formation assay: Cells were seeded in flat-bottomed twelve-well plates with 1 mL RPMI 1640 supplemented with 10% FBS. Two days later, the medium was replaced with new medium and the culture was continued for additional 12 days. Thereafter, colonies were fixated with methanol and stained with 0.05% crystal violet, then counted. The number of colonies containing 50 cells or more was counted under a microscope plate clone formation efficiency.” and colored it in red. Thanks for your carefulness.

8) The question if the other 2 binding sites (2 and 3) show a similar behavior like in Fig. 6g and h and in Suppl. Fig. 10 was not answered yet.

Answer: We appreciate the reviewer’s suggestion. We performed the ChIP-qPCR assay and confirmed the direct binding of YY1 to the miR-500a-5p -site 1, 2 and 3 proximal promoter in CRC cells (Figure 6g & h). We made necessary interpretation in RESULT section and Figure Legends in Figure 6g & h in colored in red. We have

revised the text by stating “Next, forced -expression of YY1 is up-regulated by binding of YY1 to the miR-500a-5p-site 1, 2 and 3 proximal promoter. Moreover, transient transfection of p300 decreased, whereas over-expression of HDAC2 increased, by binding of YY1 to the miR-500a-5p promoter in CRC cells (Figure 6g and Supplementary Figure 11a). In contrast, siRNA-mediated knock-down of YY1 is down-regulated by binding YY1 to the miR-500a-5p promoter and vice versa (Figure 6h and Supplementary Figure 11b).”

9) I would suggest that the authors include all their 81 patients in the analysis of Suppl. Fig. 7.

Answer: We agree with the reviewer. We performed the qPCR experiments and studied the expression relationships between miR-500a-5p and host gene CLCN5 in 81 CRC tissues. We found that the expression of miR-500a-5p was no relevant to CLCN5 ($r = -0.159$, $p = 0.157$) (Supplementary Figure 8). Thank you.

Minor points:

1) Please add the antibodies Ki-67 and CD105 in the IHC method part.

Answer: We have added the antibodies for Ki-67 and CD105 in the IHC method part. Thanks you.

2) Lane 43, 342, 355 and 437: “FK-228-treated” instead of “FK-228-induced”.

Answer: Thanks for your carefulness. We have corrected this mistake in colored in red.

3) Lane 70: “and its expression is modulated via the.. ” instead of “and modulates expression...”.

Answer: We have corrected this mistake in colored in red. Thanks you.

4) Lane 162:” the HDAC2 gene’ instead of “HDAC2 genes”.

Answer: We have corrected this mistake in colored in red. Thanks you.

5) Please revise the following lanes: 185-186; 194-195; 207-208; 228-229; 251-252; 261-262; 294-298; 310; 330; 332; 333-335; 427; 433-435; 488-489; 722-723; 729-731; 896-897; 926-928.

Answer: We appreciate the reviewer’s suggestion. We have carefully revised our paper. We try to correct some negligence in language. Thank you.

6) Lane 233: “miR-500a-5p gene” instead of miR-500a-5p in the promoter region”.

Answer: We have corrected this mistake in colored in red. Thanks you.

7) Lane 289: What do the authors mean with “ somatic cells”?

Answer: We have been deleted in a word“somatic”. Thanks you.

8) Lane 348: What is NS?

Answer: We have corrected this mistake in colored in red. Thanks.

9) Lane 429-430: Please revise.

Answer: We agree with the reviewer. We have revised the text by stating that It was previously reported that p300 is mutated in several forms of cancer suggesting a tumor suppressor role for this protein. Some reports have shown that p300 exist in multi-molecular complexes *in vivo* and function as co-activators or co-repressor for a variety of HDAC3 and YY1.^{21, 38, 39} Consistently, we showed that p300, HDAC2 formed a complex with YY1 to bind to the miR-500a-5p promoter YY1 site. Functionally, YY1 and HDAC2 inhibits miR-500a-5p promoter transcription (Figure 5c and Figure 6g & h), whereas, ectopic of expression of p300 weakened YY1 and HDAC2 to bind to the miR-500a-5p promoter YY1-binding site, and activated the miR-500a-5p promoter transcription in deed in GC cells. Therefore, we believe that p300/YY1/miR-500a-5p/HDAC2 signalling axis plays important roles cancer development” and in colored in red. Thanks you.

10) Lane 681: “4 attograms (Ag) polybrene” seems very low. Please check.

Answer: Thanks for your carefulness. We have corrected this mistake in colored in red.

11) Lane 683: The C-terminal construct of HDAC2 is not appearing in the Result section.

Answer: We have corrected this mistake in colored in red. Thanks you.

12) Lane 919: Please mention briefly the microarray; lane 477: Delete “ MicroRNA” in the headline as the paragraph describes both types of microarrays, microRNAs and mRNAs.

Answer: We have added to microarray section in colored in red. In addition, we have deleted “MicroRNA” in the headline. Thanks you.

13) Lane 899: Dunnett’s T3 multiple comparison test is not mentioned in the Method section.

Answer: We have corrected this mistake in the “Method” section in colored in red. Thanks.

14) Lane 934 and lane 31 in the Suppl. Information: exchange “micrographs” with “results”.

Answer: We have corrected this mistake in colored in red. Thanks you.

15) Lane 936 and Suppl. Information lane 33: exchange” inoculation” with “ seeding”.

Answer: We have corrected this mistake in colored in red. Thanks you.

16) Lane 947: “measured starting from 13 days” instead of “measured at thirteen days”.

Answer: We have corrected this mistake in colored in red. Thanks you.

17) Lane 966: "CRC tissues" add: "measured by qPCR".

Answer: We have corrected this mistake in colored in red. Thanks you.

18) Lane 1007: "measured starting from 11 days' instead of "measured 11 days".

Answer: We have corrected this mistake in colored in red. Thanks you.

19) Please revise Lanes 6, 17, 52 and 72 in Suppl. Information.

Answer: We have corrected this mistake in colored in red. Thanks you.

Reviewer #2 (Remarks to the Author):

The authors have resubmitted a significantly improved manuscript in which they included a substantial amount of new data answering most of my comments. However, a few of my concerns remain.

- Besides HDAC2, the authors now demonstrate that XIAP and RICTOR are miR-500a-5p targets as well. Their concluding sentence (194-195) from this data is confusing. Moreover, since XIAP and RICTOR are miR-500a-5p targets, do XIAP and RICTOR also functionally contribute to the tumour suppressive effects of miR-500a-5p in CRC cells? Or is it just HDAC2 that is involved?

Answer: We have performed experiments using western blot assays and have added this Supplementary figure 7 into Panel b. We have revised the sentence (194-195) by stating "XIAP and RICTOR were also reported as targets genes of miR-500a-5p. Our western blotting data demonstrated that the expression of XIAP or RICTOR is decreased in CRC cell lines SW620 and LoVo treated with miR-500a-5p (Supplementary Figure 7a), whereas, it is increased in the normal human colon epithelial cells FHC and NCM460 transfected with miR-500a-5p inhibitor (Supplementary Figure 7b). Our results suggested that XIAP, RICTOR and HDAC2 either independently or cooperatively functionally contribute to the tumour suppressive effects of miR-500a-5p in CRC cells." in colored in red. It is a practical suggestion.

- Following my request, the authors performed mass spectrometry to find additional HDAC2 interaction partners. Remarkably, within the 190 potential interaction partners, YY1 and p300 were not identified. The authors should comment on this.

Answer: This is a good point. Our manuscript previously showed that 190 protein potential interaction partners of HDAC2 were identified (Supplementary table 5), which YY1 and p300 were not identified, using mass spectrometric (MS) analyses. We previously described a detailed protocol for the mass spectrometric analysis in the Materials and Methods section.

1. The lysates of LoVo cells were incubated with 3 µg HDAC2 antibody for 3 h at 4 °C followed by incubation with the precleared protein A/G-agarose bead slurry.
2. The protein samples was collected and dialyzed overnight at 4 °C.

3. Samples were analyzed on a Thermo Scientific Q Exactive mass spectrometer.....

In the revised version, we performed additional experiments for SDS-PAGE. Gel electrophoresis of proteins used separate and selected gel sections were excised. Peptides were analyzed HPLC-MS/MS. We found that 306 proteins including YY1 and p300 potential interaction partners of HDAC2 were identified. We replaced the old supplementary Table 5 with new one.

We have made necessary revision in the section of “Materials and methods” by stating “The lysates of LoVo cells were incubated with 3 mg HDAC2 antibody for 3 h at 4 °C followed by incubation with the precleared protein A/G-agarose bead (Roche, Mannheim, Germany) slurry. Proteins samples were visualized following SDS-PAGE using a colloidal Coomassie blue stain (Invitrogen) according to the manufacturer's specifications. Selected gel sections were excised, destained in methanol/H₂O, and digested in-gel with L-(tosylamido-2-phenyl) ethyl chloromethyl ketone-modified trypsin (Promega; Madison, WI). Trypsin was added (in 50 mM ammonium bicarbonate) in an approximately 1 to 25 ratio (enzyme to protein), and in-gel digestion was allowed to continue overnight (37°C). Peptides were extracted from the gel slices into a 50% acetonitrile solution. Peptides were analyzed on a Thermo Scientific Q Exactive mass spectrometer (Fitgene Biotechnology CO., LTD, Guangzhou, China).” in colored in red. Thanks you!

- Fig. 1E, the miRNA ISH images are still not clear. The more intense “signal” in healthy tissue might as well be caused by differences in counter stain intensity. They should therefore provide images of miRNA ISH done on healthy and neighbouring tumour tissue on the same slide.

Answer: It is a practical suggestion. We have done extra ISH experiments and found that miR-500a-5p-positive signals were expressed in the cells of tissues adjacent to carcinoma of colon. On the contrary, colon carcinoma tissues did not express miR-500a-5p as exemplified in Figure 1e and Supplementary Figure 1. Thank you!

- There are two Luciferase assay descriptions in Methods section.

Answer: We have revised the text in the Materials and Methods section in colored in red. It is a practical suggestion.

- Line 110: miR-5001-5p should be miR-500a-5p.

Answer: We have corrected this mistake. Thanks you.

REVIEWERS' COMMENTS:

Reviewer #1 (Remarks to the Author):

The second revised version of the manuscript "The p300/YY1/miR-500a-5p/HDAC2 signalling axis regulates cell proliferation in human colorectal cancer" shows again an improvement. From my point of view there are mostly only minor points that need to be addressed before acceptance. The most important thing to improve is the English writing in some parts of the manuscript. But the authors should also give access to the mRNA microarray. So far I understood that they are having only one accession number for the microRNA microarray. The authors should also provide a link of the microarray data to the reviewers.

Here some corrections but the authors should perform in general a careful revision of the English language:

- 1) Lane 42: delete "a"
- 2) Control the format of the Supplementary Table 5
- 3) Figure 3f: explain the abbreviation HPF in Figure legend
- 4) Lane 611-612: revise, explain abbreviation HPF
- 5) Lane 188: d e f instead of D E F
- 6) Supplementary Figure 6b, right panel: correct the position of the text part
- 7) Lane 215: e instead of E
- 8) Lane 232: "gene is located" (add "is")
- 9) Lane 233: "its host gene" (add "its")
- 10) Lane 234: related instead of relevant
- 11) Lane 250: delete "p" in red
- 12) Lane 256: "was used as background" instead of "was as the background"
- 13) Lane 256: "obtained with control IgG" (add with)
- 14) Lane 265: (add of) "not only of the mature form but also of the precursor (pre-miR-500a-5p) or primary transcript (pri-miR-500a-5p) of miR-500a-5p"
- 15) Lane 299: confirm, not confirmed
- 16) Lane 301: delete "prepared"
- 17) Lane 303: "indicates" instead of "indicating"
- 18) Lane 309: correct miR-500a-5p
- 19) Lane 335-342: revise
- 20) Lane 382: down-regulated, not up-regulated
- 21) Lane 431-441: revise; what is "a variety of HDAC3 and YY1"?
- 22) Lane 446: change into: "inhibitor, FK-228, induced"
- 23) Lane 449: FK-228-induced (not treated in this case)
- 24) Lane 499-501: revise
- 25) Lane 610: fixed, not fixated
- 26) Lane 610-612: revise
- 27) Lane 632: "as per the manufacturer's instructions", revise
- 28) Lane 635: change into: Cells in each well were transfected with 0.8 µg pGL3 basic vector or the pGL3 vector harbouring the various miR-500-5p promoter regions using Lipofectamine. 2000 reagent. (1ml lipofectamine is probably an error!)
- 29) Revise lanes 638-640
- 30) Lane 773-775: revise
- 31) Lane 780: Kaplan_Meier
- 32) Figure 7: FK-228 induced (not treated)
- 33) Supplementary Figure 1: MiR-500a-5p expression was detected in CRC and adjacent...
- 34) Supplementary Figure 8: You probably used more than 20 CRC tissues! (81?)
- 35) Supplementary Figure 9: delete "does not contain any promoter sequence"
- 36) Lane 65, Suppl: cells measured by qPCR (add measured)
- 37) Supplementary Figure 10: qPCR, not q-RT-PCR
- 38) Supplementary Figure 11 a and b: you might want to change it into: Analysis of YY1 binding to various miR-500a-5p promoter regions after overexpression of YY1, HDAC2 or p300.

Reviewer #2 (Remarks to the Author):

The authors have satisfactorily addressed my questions.

REVIEWERS' COMMENTS:

Reviewer #1 (Remarks to the Author):

The second revised version of the manuscript “The p300/YY1/miR-500a-5p/HDAC2 signalling axis regulates cell proliferation in human colorectal cancer” shows again an improvement. From my point of view there are mostly only minor points that need to be addressed before acceptance. The most important thing to improve is the English writing in some parts of the manuscript. But the authors should also give access to the mRNA microarray. So far I understood that they are having only one accession number for the microRNA microarray. The authors should also provide a link of the microarray data to the reviewers.

Answer: We appreciate the reviewer’s suggestion. We give access number No: GSE115108 (Reviewer accession: [udilgsimlvifxyt](https://www.ncbi.nlm.nih.gov/geo/)) and GSE122884 (Reviewer accession: [yxwlacclbuzxox](https://www.ncbi.nlm.nih.gov/geo/)) (<https://www.ncbi.nlm.nih.gov/geo/>). Thanks.

Here some corrections but the authors should perform in general a careful revision of the English language:

1) Lane 42: delete “a”

Answer: We have deleted the word “a”. Thank you.

2) Control the format of the Supplementary Table 5

Answer: We have corrected this mistake to the Supplementary Table 5.

3) Figure 3f: explain the abbreviation HPF in Figure legend

Answer: We have explained the abbreviation HPF (High power field) accordingly.

4) Lane 611-612: revise, explain abbreviation HPF

Answer: We have explained the abbreviation HPF (High power field) accordingly. Thank you!

5) Lane 188: d e f instead of D E F

Answer: We have corrected this mistake in colored in red. Thank you

6) Supplementary Figure 6b, right panel: correct the position of the text part

Answer: We have corrected this mistake. Thank you.

7) Lane 215: e instead of E

Answer: We have corrected this mistake in colored in red. Thank you

8) Lane 232: “gene is located” (add “is”)

Answer: Thanks for your carefulness. We have corrected this mistake in colored in red. Thank you

9) Lane 233: “its host gene” (add “its”)

Answer: We have corrected this mistake in colored in red. Thank you

10) Lane 234: related instead of relevant

Answer: We have corrected this mistake in colored in red. Thanks for your carefulness. Thanks you

11) Lane 250: delete “p” in red

Answer: We have corrected this mistake in colored in red. Thank you

12) Lane 256: “was used as background” instead of “was as the background”

Answer: We have corrected this mistake in colored in red. Thank you

13) Lane 256: “obtained with control IgG” (add with)

Answer: We have corrected this mistake in colored in red. Thank you

14) Lane 265: (add of) “not only of the mature form but also of the precursor (pre-miR-500a-5p) or primary transcript (pri-miR-500a-5p) of miR-500a-5p”

Answer: We have corrected this mistake in colored in red. Thank you.

15) Lane 299: confirm, not confirmed

Answer: We have corrected this mistake in colored in red. Thank you

16) Lane 301: delete “prepared”

Answer: We have deleted the word “prepared”. Thank you.

17) Lane 303: “indicates” instead of “indicating”

Answer: We have corrected this mistake in colored in red. Thank you

18) Lane 309: correct miR-500a-5p

Answer: We have corrected this mistake in colored in red. Thank you

19) Lane 335-342: revise

Answer: We have revised the text in the section of “Results” by stating that “Next, overexpression of YY1 increased the binding of YY1 to the miR-500a-5p-site 1, 2 and 3 proximal promoter. Moreover, transient transfection of p300 decreased, whereas over-expression of HDAC2 increased, the binding of YY1 to the miR-500a-5p promoter in CRC cells (Fig. 6g and Supplementary Fig. 11a). In contrast, siRNA-mediated knock-down of YY1 decreased the binding YY1 to the miR-500a-5p promoter and vice versa (Fig. 6h and

Supplementary Fig. 11b). In addition, ectopic expression of p300 weakened HDAC2 to bind to the miR-500a-5p promoter YY1-binding sites in CRC cells (Supplementary Fig. 11c & d).” and colored in red. Thank you.

20) Lane 382: down-regulated, not up-regulated

Answer: We have corrected this mistake in colored in red. Thank you

21) Lane 431-441: revise; what is “a variety of HDAC3 and YY1”?

Answer: We have revised the text in the section of “Discussion” by stating that “It was previously reported that p300 is mutated in several forms of cancer suggesting a tumor suppressor role for this protein and provides co-repressor function. Some reports have shown that p300 exists in multi-molecular complexes in vivo and function as co-activators or co-repressor **for a variety of genes**.^{21, 38, 39} Consistently, we showed that p300 cooperates with YY1 and HDAC2 protein and function as co-repressor. Transcription factor YY1 and HDAC2 inhibit miR-500a-5p promoter transcription. Moreover, ectopic of expression of p300 weakened YY1 and HDAC2 to bind to the miR-500a-5p promoter YY1-binding sites; thus activated the miR-500a-5p promoter transcription in deed in CRC cells. Therefore, we believe that p300/YY1/miR-500a-5p/HDAC2 signalling axis plays important roles in cancer development.” and colored in red.

22) Lane 446: change into: “inhibitor, FK-228, induced”

Answer: We have corrected this mistake in colored in red. Thank you.

23) Lane 449: FK-228-induced (not treated in this case)

Answer: We have corrected this mistake in colored in red. Thank you.

24) Lane 499-501: revise

Answer: We agree with the reviewer. We have revised the text by stating that “The genes exhibit differential expression patterns between the LoVo/ miR-500a-5p cells and i-NC cells and the number of gene have had an absolute fold change greater than 1.2. Raw and processed data from the microarray were deposited in NCBI's GEO database under the accession number: GSE122884.” in colored in red. Thank you.

25) Lane 610: fixed, not fixated

Answer: We have corrected this mistake in colored in red. Thank you.

26) Lane 610-612: revise

Answer: We appreciate the reviewer’s suggestion. We have revised the text by stating that “Thereafter, colonies were fixed with methanol and stained with 0.05% crystal violet. The colonies were counted directly under a Zeiss microscope. ” in colored in red. Thank you.

27) Lane 632: “as per the manufacturer’s instructions”, revise

Answer: We have deleted the sentence “as per the manufacturer’s instructions”. Thank you.

28) Lane 635: change into: Cells in each well were transfected with 0.8 µg pGL3 basic vector or the pGL3 vector harbouring the various miR-500-5p promoter regions using Lipofectamine. 2000 reagent. (1ml lipofectamine is probably an error!)

Answer: The reviewer is very professional. We have corrected this mistake in colored in red. Thank you.

29) Revise lanes 638-640

Answer: We agree with the reviewer. We have revised the text by stating that “To examine the relationship between YY1 and miR-500a-5p promoter activity, miR-500a-5p-site 1, 2 or 3 reporter plasmid was co-transfected with YY1 or vector.” in colored in red. Thank you.

30) Lane 773-775: revise

Answer: We agree with the reviewer. We have revised the text by stating that “Comparisons between results from different groups were performed using One-way ANOVA and Dunnett’s T3 multiple comparison test.” in colored in red. Thank you.

31) Lane 780: Kaplan_Meier

Answer: We have corrected this mistake in colored in red. Thank you.

32) Figure 7: FK-228 induced (not treated)

Answer: We have corrected this mistake in colored in red. Thank you.

33) Supplementary Figure 1: MiR-500a-5p expression was detected in CRC and adjacent...

Answer: The reviewer is very professional. We have corrected this mistake in colored in red. Thank you.

34) Supplementary Figure 8: You probably used more than 20 CRC tissues! (81?)

Answer: We have corrected this mistake in colored in red. Thank you.

35) Supplementary Figure 9: delete “does not contain any promoter sequence”

Answer: Thanks for your carefulness. We have corrected this mistake in colored in red. Thank you.

36) Lane 65, Suppl: cells measured by qPCR (add measured)

Answer: We have corrected this mistake in colored in red. Thank you.

37) Supplementary Figure 10: qPCR, not q-RT-PCR

Answer: We have corrected this mistake in colored in red. Thank you.

38) Supplementary Figure 11 a and b: you might want to change it into:
Analysis of YY1 binding to various miR-500a-5p promoter regions after
overexpression of YY1, HDAC2 or p300.

Answer: We appreciate the reviewer's suggestion. We have corrected this mistake in colored in red. Thank you.